# A Robust Exact Algorithm for the Euclidean Bipartite Matching Problem

**Akshaykumar G. Gattani**[1]**, Sharath Raghvendra**[1]**, and Pouyan Shirzadian**[1]

[1]Department of Computer Science, Virginia Tech

## Abstract

Algorithms for the minimum-cost bipartite matching can be used to estimate Wasserstein distance between two distributions. Given two sets $A$ and $B$ of $n$ points in a 2-dimensional Euclidean space, one can use a fast implementation of the Hungarian method to compute a minimum-cost bipartite matching of $A$ and $B$ in $\tilde{O}(n^2)$ time. Let $\Delta$ be the spread, i.e., the ratio of the distance of the farthest to the closest pair of points in $A \cup B$. In this paper, we present a new algorithm to compute a minimum-cost bipartite matching of $A$ and $B$ with a similar worst-case execution time of $\tilde{O}(n^2 \log \Delta)$. However, when $A$ and $B$ are drawn independently and identically from a fixed distribution that is not known to the algorithm, the execution time of our algorithm is, in expectation, $\tilde{O}(n^{7/4} \log \Delta)$.

To the best of our knowledge, our algorithm is the first one to achieve a sub-quadratic execution time even for stochastic point sets with real-valued coordinates. Our algorithm extends to any dimension $d$, where it runs in $\tilde{O}(n^{2-\frac{1}{2d}} \Phi(n))$ time for stochastic point sets $A$ and $B$; here $\Phi(n)$ is the query/update time of a dynamic weighted nearest neighbor data structure. Our algorithm can be seen as a careful adaptation of the Hungarian method in the geometric divide-and-conquer framework.

## 1 Introduction

Given two distributions $\mu$ and $\nu$ defined on the sets $S_\mu, S_\nu \subseteq \mathbb{R}^d$, respectively, and an integer $p \geq 1$, the $p$-Wasserstein distance between $\mu$ and $\nu$ is the minimum cost required to transport mass from one to the other. More formally, let $\Gamma(\mu, \nu)$ be the set of all probability measures on $S_\mu \times S_\nu$ with marginal distributions $\mu$ and $\nu$. The $p$-*Wasserstein distance* between $\mu$ and $\nu$ is defined as

$$W_p(\mu, \nu) = \left( \inf_{\gamma \in \Gamma(\mu, \nu)} \int_{S_\mu \times S_\nu} \|s_1 - s_2\|^p d\gamma(s_1, s_2) \right)^{1/p},$$

where $\|s_1 - s_2\|$ denotes the Euclidean distance of $s_1$ and $s_2$.

In many applications, the distribution $\mu$ (resp. $\nu$) may not be known, but one may have access to $n$ i.i.d samples $A$ (resp. $B$) from $\mu$ (resp. $\nu$). The *empirical distribution* $\mu_n$ (resp. $\nu_n$) is a discrete distribution with $A$ (resp. $B$) as support, where each sample has a mass of $1/n$. The empirical $p$-Wasserstein distance is simply the $p$-Wasserstein distance between the empirical distributions $\mu_n$ and $\nu_n$. As $n \to \infty$, the empirical $p$-Wasserstein distance between $\mu$ and $\nu$ converges to the real $p$-Wasserstein distance between them [7, 15, 16, 46] with several results showing sharp upper and lower bounds on the empirical $p$-Wasserstein distances [10, 17, 19, 45, 49]. See also [36]

---

*Following convention from Theoretical Computer Science, all authors are ordered alphabetically.

37th Conference on Neural Information Processing Systems (NeurIPS 2023).

for a survey on such results. Due to these properties, empirical $p$-Wasserstein distance has found applications in training generative adversarial networks [8, 9, 24, 33–35], image retrieval [39, 42], graph predictions [27, 37], clustering stability validation [18, 31], and two-sample tests [13, 14, 23, 25, 41]. The convergence of the empirical distribution to the real distribution, however, exhibits the "curse of dimensionality"; that is, the convergence rate decreases as the dimension increases [16, 49]. The empirical $p$-Wasserstein distance therefore is most useful in low-dimensional settings.

Computing the empirical $p$-Wasserstein distance can be done by solving an instance of the minimum-cost bipartite matching problem as follows. Consider the complete bipartite graph on $A$ and $B$ where each edge from $a \in A$ to $b \in B$ has a cost of $\|a - b\|^p$. A *bipartite matching* (or simply a *matching*) $M$ on $A \cup B$ is a set of vertex-disjoint edges of $\mathcal{G}(A, B)$. The matching $M$ is said to be a *perfect matching* if $|M| = n$. We define the cost of a matching $M$, denoted by $w_p(M)$, as $w_p(M) = \sum_{(a,b) \in M} \|a - b\|^p$. The *pth power Euclidean bipartite matching* is a perfect matching $M_p^*$ with the minimum-cost under $w_p(\cdot)$. The empirical $p$-Wasserstein distance between $\mu$ and $\nu$ is $W_p(\mu_n, \nu_n) = \left(\frac{1}{n} w_p(M_p^*)\right)^{1/p}$.

The minimum-cost matching $M_p^*$ and therefore, the empirical $p$-Wasserstein distance can be computed in $O(n^3)$ time using the Hungarian algorithm [28]. In this paper, we adapt the Hungarian algorithm in a geometric divide-and-conquer framework. The execution time of our algorithm is similar to that of the Hungarian algorithm (Remark 3.2). However, when the input point sets $A$ and $B$ are samples drawn from the same distribution, the algorithm is asymptotically faster than the Hungarian algorithm.

**Related Work.** Computing an optimal minimum-cost bipartite matching can be done in $O(n^3)$ time using the classical Hungarian algorithm [28]. In geometric settings, the efficiency of the Hungarian algorithm can be improved to $\tilde{O}(n^2 \Phi(n))$, where $\Phi(n)$ is the query and update time of a dynamic weighted nearest neighbor data structure [3, 44, 48]. For two dimensions, $\Phi(n) = \log^{O(1)} n$ and therefore, we get a $\tilde{O}(n^2)$ time exact algorithm. For higher dimensions, however, this leads to only slightly sub-cubic execution time.

For graphs with $n$ vertices and $m$ edges, there are faster algorithms that compute an optimal minimum-cost matching [11, 21, 22] provided the edge costs are integers that are bounded by $C$[1]. However, in our setting, the coordinates of the input points are real-valued and the Euclidean distances, due to the presence of a square-root, can be irrational. As a result, these algorithms only provide an approximate solution and the Hungarian algorithm and its fast implementations remain the only known exact algorithm for the problem.

For 2-dimensional points with integer coordinates bounded by $\Gamma$, Sharathkumar [43] presented a *weakly polynomial* $\tilde{O}(n^{3/2} \log \Gamma)$ time algorithm to compute the exact 1-Wasserstein distance. The main idea was to use the approximation algorithm by Sharathkumar and Agarwal [44] to initially find a planar sub-graph that traps all the edges of the optimal matching. Then, they use the planar separator based algorithm by Lipton and Tarjan [32] to find a minimum-cost matching in this sub-graph. This algorithm, however, is restricted only to two dimensions and for $p = 1$. The design of a faster algorithm to compute the $p$th power Euclidean bipartite matching for $d$-dimensional point sets remains a major open problem. We would also note the extensive work on the design of approximation algorithms for the 1-Wasserstein distance [2, 4–6, 12, 20, 26, 30, 38, 40] as well as the 2-Wasserstein distance [1, 29].

**Our Results.** For any point set $P$ in the Euclidean space, the *spread* of $P$ is the ratio of the distance of the farthest pair to the closest pair in $P$. The main result of this paper is presented in Theorem 1.1.

**Theorem 1.1.** *There exists a randomized algorithm that, given any two point sets $A$ and $B$ sampled independently and identically from a distribution $\mu$ inside the unit $d$-dimensional hypercube, where $|A| = |B| = n$ and $\mu$ is not known to the algorithm, computes an exact Euclidean bipartite matching between $A$ and $B$ and has an expected running time of $\tilde{O}(n^{2 - \frac{1}{2d}} \Phi(n) \log \Delta)$; here, $\Delta$ is the spread of the points in $A \cup B$.*

When $A$ and $B$ are i.i.d samples from a distribution $\mu$ in the unit square ($d = 2$) and $\mu$ is not known to the algorithm, our algorithm achieves a weakly polynomial sub-quadratic execution time of $\tilde{O}(n^{7/4} \log \Delta)$. Our algorithm easily extends to any constant $p > 1$ and any fixed $d$-dimensional

---

[1]The execution time of these algorithms have a dependence on $\log C$, making them weakly polynomial.

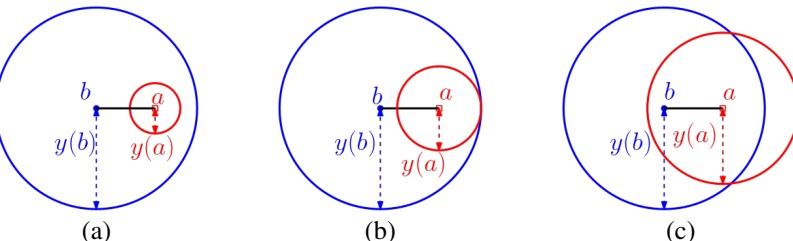

Figure 1: (a) An infeasible edge, (b) a feasible matching edge satisfying constraint (2), (c) a feasible non-matching edge satisfying constraint (1).

space (we present our result for any arbitrary $p$ as well as the proof of all our claims in the Appendix). For instance, for $p = 2$ and $d = 2$ as well as for $p = 1$ and $d = 3$, the execution time of our algorithm can be bounded by $\tilde{O}(n^{11/6}\Phi(n)\log\Delta)$. However, for simplicity in exposition of the ideas, we restrict the presentation of our algorithm to two-dimensional ($d = 2$) and Euclidean settings ($p = 1$). To the best of our knowledge, this is the first sub-quadratic weakly polynomial exact algorithm for computing the $p$th power Euclidean bipartite matching for 2-dimensional stochastic point sets. Furthermore, for many distributions, such as the uniform distribution on the unit square, one can show that, with high probability, the spread of the point set is bounded by a polynomial in $n$. For all such distributions, we achieve a strongly polynomial sub-quadratic exact algorithm.

Our algorithm can be seen as an elegant adaptation of the classical Hungarian algorithm in a quadtree based divide-and-conquer framework. The previous speed-ups of the Hungarian algorithm in geometric settings are based on sophisticated data structures that are hard to implement. In contrast, our geometric improvements are relatively simple, which allows us to implement and compare our algorithm with the standard implementation of the Hungarian algorithm. Experiments suggest that our algorithm outperforms the Hungarian algorithm for synthetic data samples drawn from two (possibly different) distributions $\mu$ and $\nu$ as long as the 1-Wasserstein distance between $\mu$ and $\nu$ is small. Our algorithm also outperforms the Hungarian algorithm on data samples drawn from the New York Taxi dataset, where one sample is drawn from the set of request pick-up locations ($\mu$) and the other sample is drawn from the request drop-off locations ($\nu$). Experiments also suggest that our analysis may not be tight, at least for the case where $\mu = \nu$ is the uniform distribution inside the unit square. Obtaining a tighter analysis of our algorithm remains an important open question. Next, we present an overview of our divide-and-conquer algorithm.

**Our Approach.** The classical Hungarian algorithm is based on the primal-dual framework where, along with a matching $M$, for every point $v \in A \cup B$, the algorithm maintains a non-negative *dual weight* $y(v)$. The matching $M$ and dual weights $y(\cdot)$ are *feasible* if, for every edge $(a, b) \in A \times B$,

$$y(b) - y(a) \le \|a - b\|, \qquad (a, b) \notin M, \qquad (1)$$
$$y(b) - y(a) = \|a - b\|, \qquad (a, b) \in M. \qquad (2)$$

The Hungarian algorithm initializes $M$ to be an empty matching and the dual weights of every vertex $v \in A \cup B$ to be 0. It then incrementally builds a perfect matching $M$ while maintaining its feasibility and returns the constructed feasible perfect matching. It is well-known that a perfect matching that is feasible is also a minimum-cost matching.

In the Euclidean setting, we interpret the dual weight $y(v)$ of any point $v$ as the radius of a disc centered around $v$. We refer to these discs as the *dual discs*. For any edge $(a, b)$, if the dual disc of $a$ is in the interior of the dual disc of $b$ (Figure 1(a)), then $y(b) - y(a) > \|a - b\|$, making the dual assignment infeasible. Therefore, for any feasible dual weight assignment, the dual disc of $a$ cannot be completely inside the dual disc of $b$. The condition (2) corresponds to Figure 1(b) where the boundary of the disc of $a$ touches the boundary of the disc of $b$ from inside. Figure 1(c) is an example of an edge that only satisfies (1).

In order to implement the Hungarian algorithm in the geometric divide-and-conquer framework, we can use a quadtree $\mathcal{Q}$ with the unit square as the root node. Each node of this tree represents a square $\square$ and if $\square$ has more than one input point, then it has four children that are obtained by splitting $\square$ into four equal squares. Squares with no more than one point become a leaf node in this tree.

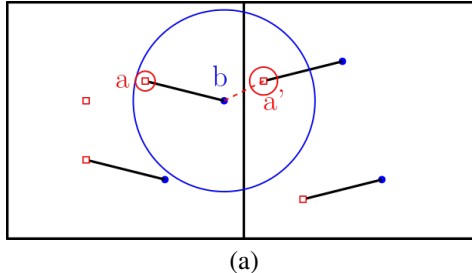 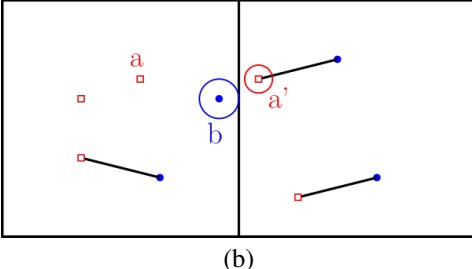

(a)                                                                (b)

Figure 2: (a) Feasible matchings for children that is infeasible when combined at the parent (the edge $(b, a')$ is infeasible), (b) Restricted feasible matchings inside the children, which is also restricted feasible when combined at parent.

Consider the following naïve implementation of the Hungarian algorithm in a quadtree based geometric divide-and-conquer framework. Given a square $\square$ of the quadtree $\mathcal{Q}$, we recursively find a feasible matching in each of the four children. Doing so, however, may lead to a dual weight assignment that violates the feasibility condition for *bridge edges*, i.e., edges that cross the boundary of the children squares. See Figure 2(a) that illustrates a dual feasible assignment at two children squares, which when combined causes the edge $(a', b)$ to violate the feasibility condition (1). Thus, implementing the conquer step becomes challenging since we cannot simply combine the feasible dual assignments at the children to achieve a feasible dual assignment at the parent.

To overcome this difficulty, we add an additional condition that restricts the dual disc of a point $v$ inside any child square $\square'$ to always stay inside $\square'$, i.e., $y(v) \leq d(v, \square')$, where $d(v, \square')$ is the Euclidean distance of $v$ to the boundary of $\square'$. Figure 2(b) shows a restricted dual assignment at the two children squares, which remains feasible when combined at the parent. Such a *restricted feasible matching* inside each of the four children squares of $\square$ trivially combine to form a feasible matching at $\square$. Thus, one can apply the divide-and-conquer framework to compute such a restricted feasible matching. Unfortunately, however, the restricted feasible matchings computed at the children may be of an extremely low cardinality. This is particularly true when most of the input points are close to the boundaries of the children squares. See, for instance, in Figure 3(a), the restricted feasible matching is an empty one since most of the blue points are close to the boundary. As a result, the divide step may not make any progress in computing a perfect matching and the conquer step may have to do significant amount of work to find a perfect matching, causing the execution time of the conquer step to be the same as that of the standard implementation of the Hungarian algorithm.

We observe that if the quadtree $\mathcal{Q}$ is constructed using a *random-shift*, then, in expectation, only sub-linearly many points are close to the boundary of any square of $\mathcal{Q}$. By combining this with the fact that the average length of a matching edge for $n$ points drawn from an arbitrary distribution $\mu$ inside unit square is $\approx 1/\sqrt{n}$, we are able to bound the expected number of unmatched points in the restricted matchings by $\tilde{O}(n^{3/4})$, leading to an execution time of $\tilde{O}(n^{7/4})$ for the conquer steps across all cells at each level of the quadtree and an overall execution time of $\tilde{O}(n^{7/4} \log \Delta)$.

## 2   Preliminaries

We begin by introducing the notations necessary to describe our algorithm. Given $A \cup B$ and any square $\square$, let $A_\square = A \cap \square$, $B_\square = B \cap \square$, and $n_\square = |A_\square \cup B_\square|$. Let $\ell_\square$ denote the side-length of $\square$. For any point $v \in A_\square \cup B_\square$ inside $\square$, let $d(v, \square)$ denote the Euclidean distance of $v$ to its closest point on the boundary of $\square$. We say that $v$ is $\delta$-close to $\square$ if $d(v, \square) \leq \delta \ell_\square$. Let $n_\square^\delta$ denote the number of points of $A_\square \cup B_\square$ that are $\delta$-close to $\square$.

**Randomly Shifted Quadtree.** Given the input points $A \cup B$ inside a unit square, a *randomly shifted quadtree* on the input can be constructed as follows. Let $\xi$ be a point chosen uniformly at random from the unit square $[0, 1]^2$. Define the square $\square^* := [-4, 4]^2 + \xi$ to be the root of the quadtree $\mathcal{Q}$. Recursively construct $\mathcal{Q}$ by decomposing any square $\square$ with $n_\square > 1$ into four equal squares, each of which become the children of $\square$. Any square with exactly one point becomes a leaf square of the quadtree. Given that the spread of the points in $A \cup B$ is $\Delta$, the height of $\mathcal{Q}$ is $O(\log \Delta)$.

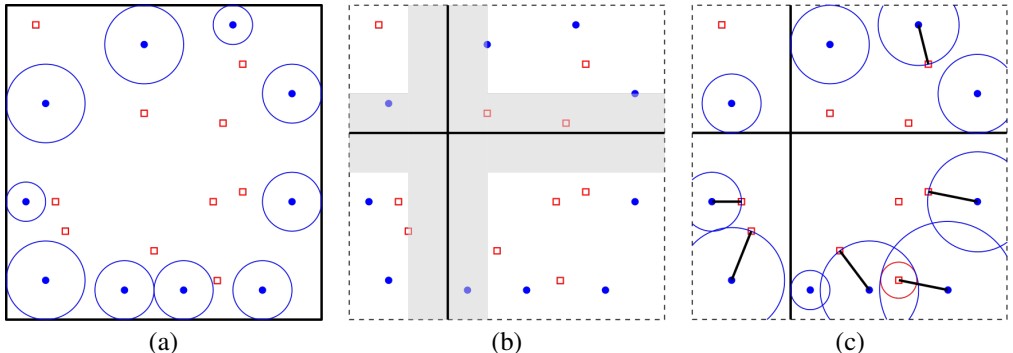

(a)                         (b)                         (c)

Figure 3: (a) A □-MCM where all points of $B$ (blue points) are unmatched, (b) the boundaries of the cells of the randomly shifted quadtree (solid line) and the $\delta$-close points to the cells of the quadtree (points in the shaded area), (c) the matchings constrained to the cells of the quadtree.

For any square □ of the quadtree $\mathcal{Q}$, in the next lemma, we show that due to the random shift, there are not many points that are very close to the boundary of □.

**Lemma 2.1.** *For any square □ of a randomly shifted quadtree and any $\delta \in (0, 1/2)$, $\mathbb{E}\left[n_\square^\delta\right] = O(\delta \mathbb{E}\left[n_\square\right])$.*

**Constrained Matching.** For any square □, we introduce a variant of the minimum-cost matching problem, which we refer to by the *minimum-cost □-constrained matching* problem. Given a square □, consider a matching $M_\square$ of the points in $A_\square \cup B_\square$. Any point $v \in A_\square \cup B_\square$ is a *free* point with respect to $M_\square$ if $v$ is not matched in $M_\square$. Let $A_\square^F$ (resp. $B_\square^F$) denote the set of free points of $A_\square$ (resp. $B_\square$) with respect to $M_\square$. We define the □-*constrained cost* of the matching $M_\square$, denoted by $w_\square(M_\square)$, as

$$w_\square(M_\square) := \sum_{(a,b) \in M_\square} \|a - b\| + \sum_{b \in B_\square^F} \mathrm{d}(b, \square). \tag{3}$$

For any square □, the *minimum-cost □-constrained matching* (□-*MCM*), denoted by $M_\square^*$, is simply a matching with the minimum □-constrained cost. In Lemma 2.2, we show that for the root square $\square^*$ of $\mathcal{Q}$, the minimum-cost $\square^*$-constrained matching is also a minimum-cost matching on $A \cup B$.

For any arbitrary square □ of $\mathcal{Q}$, $M_\square^*$ might be of a very small cardinality. See, for instance in Figure 3(a) where $B$ is the set of blue points, Equation (3) is minimized when all blue points are free points; i.e., $w_\square(M_\square^*) = \sum_{b \in B} \mathrm{d}(b, \square)$ and $M_\square^*$ is an empty matching. Nonetheless, by using the bound on the number of $\delta$-close points to any square □ of a randomly-shifted quadtree, we can bound the expected number of free points with respect to any □-MCM $M_\square^*$. In the example of Figure 3(b), the solid lines show the boundary of the squares of a randomly-shifted quadtree, and as shown in Figure 3(c), only a few points will be free with respect to the optimal matchings constrained to the squares of the quadtree.

**Lemma 2.2.** *For any square □ of a randomly shifted quadtree and any minimum-cost □-constrained matching $M_\square^*$, (i) the expected number of free points of $B_\square$ with respect to $M_\square^*$ is $\tilde{O}\left(n^{3/4}\right)$ in 2-dimensions (and $\tilde{O}(n^{1-\frac{1}{2d}})$ in $d$ dimensions), and (ii) if □ is the root square, then $M_\square^*$ is a minimum-cost perfect matching on $A \cup B$.*

For the rest of the paper, we design a primal-dual method to compute a $\square^*$-MCM, which by Lemma 2.2 is also a minimum cost matching on $A \cup B$.

**Constrained Feasibility.** Similar to the Hungarian algorithm, we devise a primal-dual method to compute a □-MCM. For any square □ of $\mathcal{Q}$, we say that a matching $M_\square$ on $A_\square \cup B_\square$ along with a set of non-negative dual weights $y(\cdot)$ for the points in $A_\square \cup B_\square$ is □-*feasible* if,

$$y(b) - y(a) \leq \|a - b\|, \qquad \forall (a, b) \in A \times B, \tag{4}$$
$$y(b) - y(a) = \|a - b\|, \qquad \forall (a, b) \in M_\square. \tag{5}$$
$$y(b) \leq \mathrm{d}(b, \square), \qquad \forall b \in B_\square, \tag{6}$$
$$y(a) = 0, \qquad \forall a \in A_\square^F. \tag{7}$$

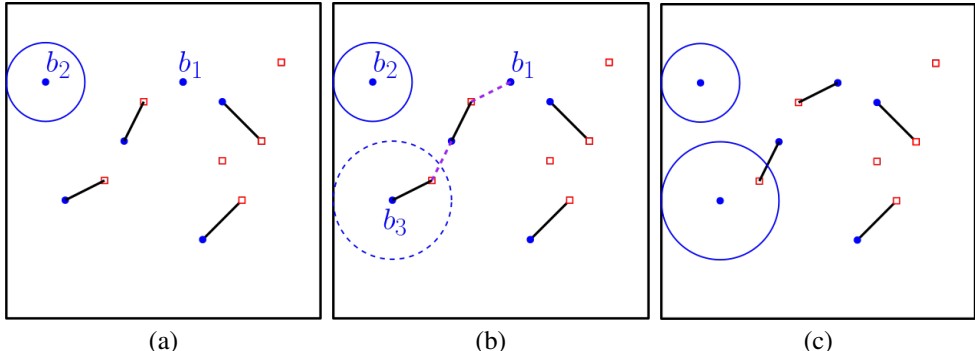

Figure 4: (a) A $\square$-feasible matching $M_\square$ where $b_1$ is a $\square$-free point and $b_2$ is a free point that is not $\square$-free, (b) an admissible path $P$ (dashed line) from $b_1$ to a point $b_3$ whose vertex slack is zero, and (c) the $\square$-optimal matching obtained from augmenting $M_\square$ along $P$.

Note that (4) and (5) are conditions identical to the ones maintained by the Hungarian algorithm. Condition (6) ensures that the dual discs stay inside the square and condition (7) ensures that the $\square$-MCM is of minimum cost.

Given a $\square$-feasible matching $M_\square, y(\cdot)$, we say that any point $b \in B_\square$ is a $\square$-*free point* with respect to $M_\square$ if $b$ is not matched in $M_\square$ and $y(b) < \mathrm{d}(b, \square)$. Note that a free point may not be $\square$-free. For instance, in Figure 4(a), the blue point $b_2$ is free but not a $\square$-free point since its dual weight is equal to its distance to $\square$. Let $\mathcal{B}_\square^F$ denote the set of all $\square$-free points in $B_\square$. We say that a $\square$-feasible matching $M_\square, y(\cdot)$ is $\square$-*optimal* if it does not have any $\square$-free points, i.e., $\mathcal{B}_\square^F = \emptyset$. The following lemma shows that any $\square$-optimal matching $M_\square, y(\cdot)$ is a $\square$-MCM. Thus, we focus on computing a $\square$-optimal matching.

**Lemma 2.3.** *Let $M_\square, y(\cdot)$ be a $\square$-optimal matching on $A_\square \cup B_\square$. Then, $M_\square$ is a minimum-cost $\square$-constrained matching.*

Given a square $\square$ of $\mathcal{Q}$ and its four children $\square_1, \square_2, \square_3,$ and $\square_4$, we can obtain a $\square$-feasible matching at $\square$ by simply combining the $\square_i$-optimal matchings from each child $\square_i$. We use this property to design our divide-and-conquer algorithm for computing $\square$-optimal matching.

**Lemma 2.4.** *For any square $\square$, let $\square_i$, $i \in [1, 4]$ be the set of all children of $\square$ and let $M_i, y(\cdot)$ denote a $\square_i$-optimal matching. Then, the matching $\bigcup_{i=1}^{4} M_i, y(\cdot)$ is a $\square$-feasible matching.*

Next, we define slack and an augmenting path with respect to a $\square$-feasible matching.

**Slacks.** For any square $\square$, any $\square$-feasible matching $M_\square, y(\cdot)$, and any pair of points $(a, b) \in A_\square \times B_\square$, we define the *edge slack* on the feasibility conditions for edge $(a, b)$, denoted by $s(a, b)$, as $s(a, b) = \|a - b\| - y(b) + y(a)$. We say that $(a, b)$ is *admissible* if $s(a, b) = 0$. Furthermore, for any point $b \in B_\square$, we define the *vertex slack* of $b$ with respect to condition (6) to be $s_\square(b) := \mathrm{d}(b, \square) - y(b)$. We refer to the edge slack (resp. vertex slack) as slack when the edge (resp. vertex) is obvious from the context.

While the definition of edge slacks are identical to what is used in the Hungarian algorithm, the definition of vertex slacks is new. Since the dual weights associated with any $\square$-feasible matching satisfies (4)–(7), both edge slacks and vertex slacks will be non-negative.

**Admissible Augmenting Path.** For any square $\square$, suppose $M_\square, y(\cdot)$ is a $\square$-feasible matching on $A_\square \cup B_\square$. An *alternating path* with respect to $M_\square$ is a simple path on $A_\square \cup B_\square$ whose edges alternate between those in $M_\square$ and those not in $M_\square$. An *admissible augmenting path* (or simply *admissible path*) is any alternating path consisting only of zero slack edges that starts at a $\square$-free point $b \in \mathcal{B}_\square^F$ and ends at either (i) a free point $a \in A_\square^F$ or (ii) a point $b' \in B_\square$ with a slack of $0$. We can *augment* a $\square$-feasible matching $M_\square$ along an admissible path $P$ by setting $M_\square \leftarrow M_\square \oplus P$, i.e., we remove all matching edges of $P$ from $M_\square$ and add all non-matching edges of $P$ to $M_\square$ (see Figure 4). The following lemma shows that augmenting a matching along an admissible path does not violate $\square$-feasibility conditions and also reduces the number of $\square$-free points of $B_\square$ by one.

**Lemma 2.5.** *Suppose $M_\square, y(\cdot)$ is a $\square$-feasible matching and $P$ is an admissible path. After augmenting $M_\square$ along $P$, the matching $M_\square, y(\cdot)$ remains $\square$-feasible. Furthermore, the augmentation reduces the number of $\square$-free points of $B_\square$ with respect to $M_\square$ by one.*

**Residual Network.** In order to assist in computing admissible paths with respect to a $\square$-feasible matching $M_\square, y(\cdot)$, we define a *residual network* of $A_\square \cup B_\square$ with respect to $M_\square$ as follows. The vertex set of the residual network is the set of points $A_\square \cup B_\square$ and a source vertex $s$. For any pair of points $(a, b) \in A_\square \times B_\square$, if $(a, b) \in M_\square$, we add an edge directed from $a$ to $b$ with a weight $s(a, b)$ to the residual network. Otherwise, if $(a, b) \notin M_\square$, we add an edge directed from $b$ to $a$ with a weight $s(a, b)$. In addition, we add zero-weight edges from the source $s$ to every $\square$-free point in $\mathcal{B}_\square^F$. Note that any zero-weight directed path from the source vertex $s$ to a free point $a \in A_\square^F$ or a zero-slack point $b \in B_\square$ in this residual network is an admissible path.

# 3 Algorithm

In this section, we describe our algorithm for computing an optimal matching on a point set $A \cup B$. Our algorithm builds a randomly shifted quadtree $\mathcal{Q}$. For any square $\square$ of the quadtree $\mathcal{Q}$, we denote the set of children of $\square$ in $\mathcal{Q}$ by $\mathsf{C}[\square]$.

We describe our divide-and-conquer algorithm with respect to an arbitrary square $\square$ in $\mathcal{Q}$. Our algorithm computes a $\square$-optimal matching $M_\square, y(\cdot)$.

**Base case.** If $\square$ is a leaf of $\mathcal{Q}$, let $v$ be the only point in $\square$. If $v \in B$, set $y(v) \leftarrow \mathrm{d}(v, \square)$ and $M_\square \leftarrow \emptyset$. If $v \in A$, set $y(v) \leftarrow 0$ and $M_\square \leftarrow \emptyset$ and return (see lines 1–4 of Algorithm 1).

**Divide step.** If $\square$ is not a leaf of $\mathcal{Q}$, for each child $\square' \in \mathsf{C}[\square]$, recursively compute the $\square'$-optimal matching $M_{\square'}, y(\cdot)$ on $A_{\square'} \cup B_{\square'}$ (see lines 6 and 7 of Algorithm 1).

**Conquer step.** For any child $\square' \in \mathsf{C}[\square]$, let $M_{\square'}, y(\cdot)$ denote the $\square'$-optimal matching returned by the algorithm. Set $M_\square := \bigcup_{\square' \in \mathsf{C}[\square]} M_{\square'}$ as the union of all matchings computed inside the children of $\square$. Let $\mathcal{B}_\square^F$ denote the $\square$-free points of $B_\square$ with respect to $M_\square$. To obtain a $\square$-optimal matching, our algorithm iteratively executes the CONSTRAINEDHUNGARIANSEARCH procedure that adjusts the dual weights to find an admissible path $P$ with respect to $M_\square$. Then, the AUGMENT procedure updates $M_\square$ by augmenting the matching along $P$. The algorithm continues to search and augment until there are no $\square$-free points of $B_\square$, i.e., until $M_\square, y(\cdot)$ is a $\square$-optimal matching. See lines 8–11 of Algorithm 1. We describe the details of the CONSTRAINEDHUNGARIANSEARCH and AUGMENT procedures below.

*1- CONSTRAINEDHUNGARIANSEARCH procedure:* Compute the residual network with respect to $M_\square$ and execute a Dijkstra's shortest path algorithm starting from the source $s$. Let $\kappa_v$ denote the shortest path distance from $s$ to $v$ as computed by the Dijkstra's algorithm. Let

$$\kappa = \min\{\min_{a \in A_\square^F} \kappa_a, \min_{b \in B_\square} \kappa_b + s_\square(b)\},$$

and let $u$ denote the point that achieves this minimum. Note that $u \in A_\square^F \cup B_\square$. Let $P_u$ denote the shortest path from $s$ to $u$ and $P$ denote the path obtained by removing $s$ from $P_u$. For any $v \in A_\square \cup B_\square$, if $\kappa_v < \kappa$, update its dual weight to $y(v) \leftarrow y(v) + \kappa - \kappa_v$.

*2- AUGMENT procedure:* Augment $M_\square$ along $P$.

Our algorithm returns the matching $M_{\square^*}$ computed at the root $\square^*$ of $\mathcal{Q}$ as the minimum-cost matching between $A$ and $B$. This completes the description of our algorithm. The pseudo-code of our divide-and-conquer algorithm is provided in Algorithm 1.

## 3.1 Proof of correctness

Recollect that for the root square $\square^*$ of $\mathcal{Q}$, from Lemma 2.2, a $\square^*$-optimal matching is a minimum-cost matching of $A$ and $B$. Therefore, it suffices to show that, at each square $\square$ of $\mathcal{Q}$, our algorithm computes a $\square$-optimal matching.

We begin by showing this for any leaf square $\square$, which by definition contains exactly one point. Since there are no edges inside $\square$, conditions (4) and (5) hold trivially. Let $v$ be the only point inside $\square$.

---

**Algorithm 1** D&CHUNGARIAN($\square, A_\square, B_\square$)

---

**Input:** A square $\square$ of quadtree $\mathcal{Q}$ and two point sets $A_\square$ and $B_\square$ inside $\square$
**Output:** A minimum-cost $\square$-constrained matching $M_\square, y(\cdot)$

1: **if** $\square$ is a leaf of $\mathcal{Q}$ **then**                                                           ▷ Base case
2:      $M_\square \leftarrow \emptyset$
3:      **//** Let $v$ be the only point in $A_\square \cup B_\square$
4:      **if** $v \in A_\square$ **then** $y(v) \leftarrow 0$ **else** $y(v) \leftarrow \mathrm{d}(v, \square)$
5: **else**
6:      **//** Let $\square_1, \square_2, \square_3,$ and $\square_4$ be the children of $\square$ in $\mathcal{Q}$              ▷ Divide step
7:      $M_{\square_i}, y \leftarrow$ D&CHUNGARIAN($\square_i, A_{\square_i}, B_{\square_i}$),     $\forall i \in [1, 4]$
8:      $M_\square \leftarrow \bigcup_{i \in [4]} M_{\square_i}$                                           ▷ Conquer step
9:      **while** $\mathcal{B}_\square^F \neq \emptyset$ **do**
10:        $P, y \leftarrow$ CONSTRAINEDHUNGARIANSEARCH($M_\square, y$)
11:        $M_\square \leftarrow$ AUGMENT($M_\square, P$)
12: **return** $M_\square, y(\cdot)$

---

If $v \in B_\square$, then $y(v) = \mathrm{d}(v, \square)$ and therefore, condition (6) holds. Otherwise, $v \in A_\square$ and our algorithm sets $y(v) = 0$; therefore, condition (7) holds. In both cases, it is easy to see that $\mathcal{B}_\square^F$ is empty and therefore the empty matching $M_\square$ and dual weight $y(v)$ is a $\square$-optimal matching.

For any non-leaf square $\square$ of $\mathcal{Q}$, at the beginning of the conquer step, the algorithm simply combines the constrained optimal matchings computed at its non-empty children. From Lemma 2.4, the resulting matching will be a $\square$-feasible matching. From Lemma 3.1 below, after the execution of the CONSTRAINEDHUNGARIANSEARCH procedure, the matching $M_\square, y(\cdot)$ remains a $\square$-feasible matching and the path $P$ returned by the procedure is an admissible path. From Lemma 2.5, the execution of AUGMENT procedure then reduces the number of $\square$-free points by one while maintaining the $\square$-feasibility conditions. Therefore, during the execution of the conquer step, the matching $M_\square, y(\cdot)$ remains $\square$-feasible and the number of $\square$-free points reduces by one at each iteration; thus, the algorithm terminates with a $\square$-optimal matching, as desired.

**Lemma 3.1.** *Given any square $\square$ of $\mathcal{Q}$ and any $\square$-feasible matching $M_\square, y(\cdot)$, after executing the* CONSTRAINEDHUNGARIANSEARCH *procedure on $\square$, the updated dual weights remain $\square$-feasible. Furthermore, the returned path $P$ is an admissible path.*

### 3.2 Proof of Efficiency

For any square $\square$, let $T_\square$ denote the execution time of the conquer step of our algorithm when executed on $\square$. Additionally, for any $i$, let $\mathcal{L}(i)$ denote the set of all squares of the quadtree that are processed at depth $i$ of recursion and let $T_i$ denote the total execution time of our algorithm across all such squares, i.e., $T_i = \sum_{\square \in \mathcal{L}(i)} T_\square$.

Recall that the conquer step of our algorithm consists of iterations, where in each iteration, it executes the CONSTRAINEDHUNGARIANSEARCH procedure followed by the AUGMENT procedure. For any square $\square$ of $\mathcal{Q}$, from Lemma 2.5, each iteration of the conquer step reduces the number of $\square$-free points of $B_\square$ by one. Therefore, the number of iterations is bounded by the number of $\square$-free points of $B_\square$, which is at most the total number of free points with respect to the $\square'$-MCM computed for all children $\square'$ of $\square$. By invoking Lemma 2.2 on all children of $\square$, we bound the expected number of iterations of the conquer step on $\square$ by $\tilde{O}(n^{3/4})$ for 2 dimensions (and $\tilde{O}(2^d n^{1 - \frac{1}{2d}})$ for $d$ dimensions).

Using any dynamic weighted nearest neighbor data structure with an update/query time of $\Phi(n_\square)$, one can execute the CONSTRAINEDHUNGARIANSEARCH procedure in $\tilde{O}(n_\square \Phi(n_\square))$ time [44, 48]. For planar point sets, $\Phi(n_\square) = \mathrm{poly}(\log n_\square)$ and therefore, each execution of the CONSTRAINEDHUNGARIANSEARCH procedure takes $\tilde{O}(n_\square)$ time. Furthermore, the AUGMENT procedure simply augments the matching along an admissible path, which can be done in $O(n_\square)$ time. Therefore, the conquer step of our algorithm executes $\tilde{O}(n^{3/4})$ iterations for 2 dimensions (and $\tilde{O}(2^d n^{1 - \frac{1}{2d}})$ iterations for $d$ dimensions), in expectation, where each iteration takes $\tilde{O}(n_\square)$ time for 2 dimensions (and $\tilde{O}(n_\square \Phi(n_\square))$ for $d$ dimensions). Since $\sum_{\square \in \mathcal{L}(i)} n_\square \leq n$, $\mathbb{E}[T_i] = \sum_{\square \in \mathcal{L}(i)} \mathbb{E}[T_\square] =$

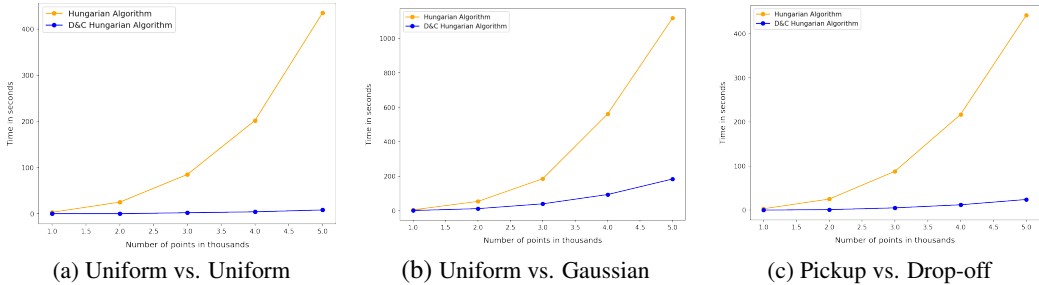

| (a) Uniform vs. Uniform | (b) Uniform vs. Gaussian | (c) Pickup vs. Drop-off |

Figure 5: The running time of our algorithm (D&C Hungarian Algorithm) and the Hungarian algorithm for computing the 1-Wasserstein distance between $\mu$ and $\nu$, where (a) both $\mu$ and $\nu$ are Uniform, (b) $\mu$ is Uniform and $\nu$ is Gaussian, and (c) $\mu$ is Pickup and $\nu$ is Drop-off.

$\tilde{O}\left(\sum_{\square \in \mathcal{L}(i)} n^{3/4} n_\square\right) = \tilde{O}(n^{7/4})$ for 2 dimensions (and $\mathbb{E}[T_i] = \tilde{O}(2^d n^{2-\frac{1}{2d}} \Phi(n))$ for $d$ dimensions). Summing over all levels of $\mathcal{Q}$, the expected execution time of our algorithm is $\tilde{O}(n^{7/4} \log \Delta)$ for 2 dimensions (and $\tilde{O}(2^d n^{2-\frac{1}{2d}} \Phi(n) \log \Delta)$ for $d$ dimensions). Using a slightly sophisticated argument which is presented in the Appendix, we can remove the $2^d$ from the execution time for the $d$-dimensional settings, leading to Theorem 1.1.

**Remark 3.2.** *When the input points $A \cup B$ are chosen arbitrarily, the number of iterations of the conquer step can be $\Theta(n_\square)$, leading to an execution time of $O(n_\square^2 \Phi(n_\square))$ per square and $O(n^2 \Phi(n) \log \Delta)$ in total.*

## 4 Experiments

In this section, we present the results of our experiments comparing the execution time of our algorithm to that of the Hungarian algorithm. Both algorithms are implemented in Java and share the same data structures[2]. For both algorithms, we use the classical implementation of Dijkstra's shortest path algorithm used in the Hungarian search procedure and do not use any weighted nearest neighbor data structures. All computations are performed using a single calculation thread on a computer with a 2.6 GHz 6-Core Intel Core i7 CPU and 16 GB of 2667 MHz DDR4 RAM.

**Datasets.** We test our algorithm on 2-dimensional synthetic and real-world datasets. For the synthetic dataset, we use two distributions, namely (i) a uniform distribution defined on the unit square (Uniform), and (ii) a Gaussian distribution constrained to the unit square with a randomly chosen mean inside the unit square and a standard deviation of $0.25$ (Gaussian). For a real-world dataset, we employ the New York Taxi dataset [47] and obtain two distributions, namely (i) the distribution of pickup locations (Pickup) and (ii) the distribution of drop-off locations (Drop-off) of passengers. We filtered the datasets by considering trips in seven dates in 2014 with (i) a trip duration of at least 3 minutes, and (ii) a trip velocity of at most $110mph$.

**Tests.** In each test, we conducted experiments using two sets of $n$ i.i.d samples from distributions $\mu$ and $\nu$ within the unit square. We executed both our divide-and-conquer algorithm and the Hungarian algorithm to compute the empirical 1-Wasserstein distance between the same set of samples and compared the execution times. In our first experiment, we used Uniform distribution for both $\mu$ and $\nu$ to compare the performance of algorithms when the distributions are the same. For the second experiment on synthetic datasets, we compared running times with different distributions - $\mu$ being Uniform and $\nu$ being Gaussian, and in the third experiment, which is on the real-world datasets, $\mu$ was set to the Pickup distribution while $\nu$ was the Drop-off distribution. In our experiments on the synthetic datasets, we recorded the number of iterations our algorithm executes at each square of the quadtree to find out how tight our analysis are. The results of our experiments are shown in Figures 5 and 6.

**Results.** As depicted in Figure 5, experiments suggest that our algorithm outperforms the Hungarian algorithm, and the improvement is most significant when the distributions are the same (Figure 5(a)). More generally, experiments suggest that our algorithm performs significantly better than the Hungar-

---

[2]Our implementations are available at https://github.com/agattani190/Exact-Euclidean-Bipartite-Matching.

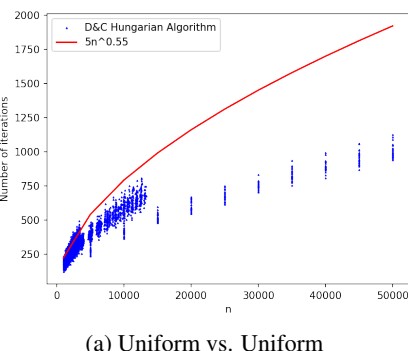
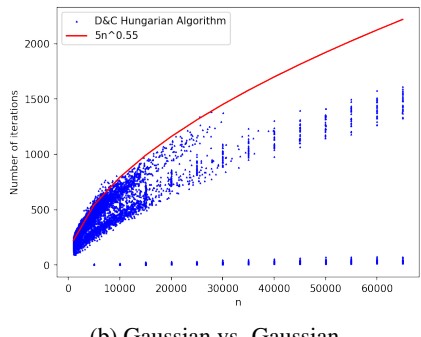

(a) Uniform vs. Uniform  (b) Gaussian vs. Gaussian

Figure 6: The number of iterations of the conquer step for a square with $n$ points when executed on samples from the same distributions ((e) Uniform and (f) Gaussian).

ian algorithm when the matching cost is low (Figures 5(a) and (c)). The reason for this improvement lies in the fact that the Hungarian algorithm performs a 'global Hungarian search' for matching each point, whereas our algorithm isolates shorter edges of the optimal matching within smaller quadtree sub-problems and execute 'local Hungarian search' for them. Therefore, while the asymptotic improvements are shown for samples from the same distribution, we expect our algorithm to perform better when the optimal matching has many edges with small cost, such as when points are drawn from two similar but not identical distributions. This is evident in our third experiment (Figure 5(c)), where we sample $n$ pick-up locations and $n$ drop-off locations from the New York Taxi dataset. Notably, pick-up and drop-off locations tend to follow different distributions, with pick-ups having a higher density around Manhattan.

Additionally, our algorithm handles larger point sets efficiently; for example, it computes the optimal solution for $50\,000$ points from the Uniform distribution in approximately 700 seconds. Finally, in Figure 6, our experiments on samples from the same synthetic dataset suggest that for a square with $n$ points, the number of iterations of the conquer step of our algorithm might be bounded by $O(n^{\alpha})$ for $\alpha \simeq 0.55$. This perhaps suggests that our upper-bound of $O(n^{3/4})$ may be an overestimate, at least for the uniform and Gaussian distributions.

## 5   Conclusion

In this paper, we adapted the classical primal-dual approaches for computing minimum-cost matching within the divide-and-conquer framework. This approach led to a sub-quadratic weakly-polynomial exact algorithm for computing minimum-cost matching on stochastic point sets. Notably, our algorithm incorporates a randomly shifted quadtree, a structure previously used only in approximation algorithms for $p$-Wasserstein distance. We conclude by highlighting some open problems for further investigation.

- An interesting question is whether it is possible to further exploit the geometry of the dual weights and give a tighter bound for the number of iterations of our algorithm at each square; thus, improving the analysis of our algorithm.

- Our analysis only requires the fact that the average length of the matching edges in large cells is small. Stochastic point sets have this property and therefore, we can achieve sub-quadratic algorithms for them. Is there a sub-quadratic exact algorithm for the case where the matching edges are long? Understanding this case may shed light into the design of sub-quadratic exact algorithm for arbitrary inputs.

- Finally, our algorithm is a weakly polynomial algorithm as its running time is dependent on the spread of the point set. Can we remove the dependence on spread while maintaining the simplicity of our algorithm?

## Acknowledgement

We would like to acknowledge Advanced Research Computing (ARC) at Virginia Tech, which provided us with the computational resources used to run the experiments. Research presented in this paper was funded by NSF CCF-1909171 and NSF CCF-2223871. We would like to thank the anonymous reviewers for their useful feedback.

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
