# Appendix of A Robust Exact Algorithm for the Euclidean Bipartite Matching Problem

**Akshaykumar G. Gattani**[1]**, Sharath Raghvendra**[1]**, and Pouyan Shirzadian**[1]

[1]Department of Computer Science, Virginia Tech

## A   Missing Proofs and Details

In this section, we present the missing proofs and details of the claims made in Sections 2 and 3.

### A.1   Properties of a Randomly Shifted Quadtree

Recall that for any square $\square$ of the randomly shifted quadtree $\mathcal{Q}$ and any parameter $\delta \in (0, 1/2)$, $n_\square$ denotes the number of points of $A \cup B$ that lie inside $\square$, among which $n_\square^\delta$ denotes the number of points that are $\delta$-close to $\square$, i.e., the number of points $u \in A_\square \cup B_\square$ inside $\square$ with $\mathrm{d}(\square, u) \le \delta\ell_\square$. Define $\mathcal{S}_\square^\delta$ to be the locus of points that are $\delta$-close to $\square$. In Figure 1, for instance, $\mathcal{S}_\square^\delta$ is shown as the green region and all 5 points inside that region are $\delta$-close to $\square$. For any geometrical shape $S$, let $A(S)$ denote the area of $S$. In this section, we show that due to the random shift of the quadtree, the expected number of points that are $\delta$-close to $\square$ is at most $O(\delta\mathbb{E}[n_\square])$, leading to Lemma 2.1.

For any point $u \in A \cup B$, let $X_u$ be an indicator random variable indicating whether $u$ lies inside $\square$ or not. More precisely, $X_u = 1$ if $u \in A_\square \cup B_\square$ and $X_u = 0$ otherwise. Similarly, let $Y_u$ be an indicator random variable such that $Y_u = 1$ if $u$ is $\delta$-close to $\square$ and $Y_u = 0$ otherwise. By definition,

$$\Pr[Y_u = 1] = \Pr[Y_u = 1, X_u = 1] = \Pr[Y_u = 1 \mid X_u = 1]\Pr[X_u = 1]. \tag{1}$$

Due to the random shift of $\mathcal{Q}$, for any point $u \in A \cup B$, if $u$ lies inside $\square$ (i.e., $X_u = 1$), then the probability that $u$ is $\delta$-close to $\square$ (i.e., $Y_u = 1$) would be the ratio of the area of $\mathcal{S}_\square^\delta$ to the area of $\square$. More precisely,

$$\Pr[Y_u = 1 \mid X_u = 1] = \frac{A(\mathcal{S}_\square^\delta)}{A(\square)} \le \frac{4\delta\ell_\square^2}{\ell_\square^2} = 4\delta. \tag{2}$$

Combining Equations (1) and (2),

$$\Pr[Y_u = 1] = \Pr[Y_u = 1 \mid X_u = 1]\Pr[X_u = 1] \le 4\delta\Pr[X_u = 1].$$

Therefore,

$$\mathbb{E}\left[n_\square^\delta\right] = \sum_{u \in A \cup B} \mathbb{E}[Y_u] = \sum_{u \in A \cup B} \Pr[Y_u = 1] \le 4\delta \sum_{u \in A \cup B} \Pr[X_u = 1] = 4\delta\mathbb{E}[n_\square].$$

**Lemma 2.1.** *For any square $\square$ of a randomly shifted quadtree and any $\delta \in (0, 1/2)$, $\mathbb{E}\left[n_\square^\delta\right] = O(\delta\mathbb{E}[n_\square])$.*

Note that when the point sets $A \cup B$ are inside the unit $d$-dimensional hypercube for any $d \ge 2$, then one can show that $\Pr[Y_u = 1 \mid X_u = 1] \le 2d\delta$, leading to the following lemma.

**Lemma A.1.** *For any $d$-dimensional hypercube $\square$ of a randomly shifted quadtree and any $\delta \in (0, 1/2)$, $\mathbb{E}\left[n_\square^\delta\right] = O(d\delta\mathbb{E}[n_\square])$.*

37th Conference on Neural Information Processing Systems (NeurIPS 2023).

## A.2  Properties of Constrained Matchings

In this section, we first show that for the root square $\square^*$ of $\mathcal{Q}$, any $\square^*$-MCM on $A \cup B$ is a minimum-cost perfect matching on $A \cup B$. Following that, given any square $\square$ of $\mathcal{Q}$ and any $\square$-MCM $M_\square$, we show that the expected $\square$-constrained cost of $M_\square$ is $O(\ell_\square \sqrt{n} \log n)$. Using this bound, we then give an upperbound on the expected number of free points with respect to $M_\square$ and conclude Lemma 2.2. In each part, we also extend our bounds to any dimension $d \geq 3$.

**Minimum-Cost Constrained Matching for the Root of Quadtree.**   For the root square $\square^*$ of $\mathcal{Q}$, suppose $M_{\square^*}$ denotes any $\square^*$-MCM on $A \cup B$ and let $M^*$ denote any minimum-cost perfect matching on $A \cup B$. In this part, we first show that $M_{\square^*}$ is a perfect matching. We then conclude that $M_{\square^*}$ is a minimum-cost perfect matching by showing that $w(M_{\square^*}) \leq w(M^*)$.

Intuitively, our construction guarantees that for any point $b \in B$, the distance of $b$ to the boundaries of $\square^*$ is more than the distance of $b$ to any point $a \in A$. In this case, any free point $b \in B^F$ will contribute a high cost of $\mathrm{d}(b, \square^*)$ to $w_{\square^*}(M_{\square^*})$ and therefore, matching $b$ to any free point of $A$ will reduce the $\square^*$-constrained cost of $M_{\square^*}$, which is a contradiction to the assumption that $M_{\square^*}$ is a $\square^*$-MCM; hence, any $\square^*$-MCM is a perfect matching on $A \cup B$. We give the details below.

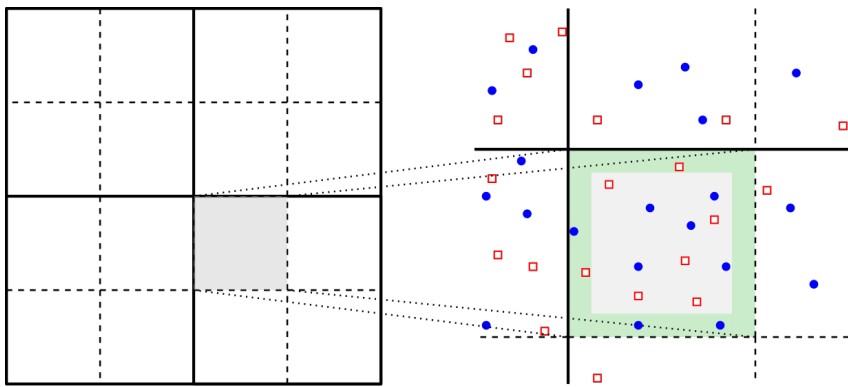

Figure 1: (left) Shows a quadtree $\mathcal{Q}$ and square $\square$ of the quadtree (shaded square), (right) shows the points of $B$ (blue disks), points of $A$ (red squares), the placement of $\square$ after randomly shifting the quadtree, and the points that are $\delta$-close to $\square$ (the points inside the green region).

For contradiction, suppose $M_{\square^*}$ is not a perfect matching and the set of free points $B^F$ is not empty. By construction, for any free point $b \in B^F$, the distance of $b$ to the boundary of $\square^*$ is at least 3, i.e., $\mathrm{d}(b, \square^*) \geq 3$. Therefore, the contribution of $b$ to $w_{\square^*}(M_{\square^*})$ is $\mathrm{d}(b, \square^*) \geq 3$. Since $|A| = |B|$, $A$ and $B$ have the same number of free points with respect to $M_{\square^*}$, i.e., $\left| A^F \right| = \left| B^F \right| > 0$. For any free point $a \in A^F$, since both $a$ and $b$ are inside the unit square, $\|a - b\| \leq \sqrt{2}$; therefore, by adding the edge $(a, b)$ to $M_{\square^*}$, the change in the $\square^*$-constrained cost of $M_{\square^*}$ would be $\|a - b\| - \mathrm{d}(b, \square^*) < 0$, i.e., adding $(a, b)$ to $M_{\square^*}$ results in a matching with a lower $\square^*$-constrained cost, which is a contradiction. Therefore, the matching $M_{\square^*}$ cannot have any free points and it is a perfect matching.

Next, we show that the cost of $M_{\square^*}$ is no more than the cost of $M^*$. By definition of the $\square^*$-constrained cost, since $M_{\square^*}$ is a perfect matching (and the set of free points of $B$ with respect to $M_{\square^*}$ is empty), $w(M_{\square^*}) = w_{\square^*}(M_{\square^*})$; similarly, since $M^*$ is a perfect matching, $w_{\square^*}(M^*) = w(M^*)$. Finally, since $M_{\square^*}$ is a $\square^*$-MCM, $w_{\square^*}(M_{\square^*}) \leq w_{\square^*}(M^*)$. Combining all three bounds,

$$w(M_{\square^*}) = w_{\square^*}(M_{\square^*}) \leq w_{\square^*}(M^*) = w(M^*). \tag{3}$$

Therefore, since $M_{\square^*}$ is a perfect matching whose cost is no more than the cost of any minimum-cost perfect matching, we conclude that $M_{\square^*}$ is also a minimum-cost perfect matching.

**Lemma A.2.** *For the root square $\square^*$ of the quadtree and any minimum-cost $\square^*$-constrained matching $M_{\square^*}$ on $A \cup B$, $M_{\square^*}$ is a minimum-cost perfect matching on $A \cup B$.*

**Expected Cost of the Constrained Matchings.**   Given any square $\square$ of $\mathcal{Q}$, let $M_\square^*$ denote any $\square$-MCM. In this part, we show that the expected $\square$-constrained cost of $M_\square^*$ is $O(\ell_\square \sqrt{n} \log n)$. Define

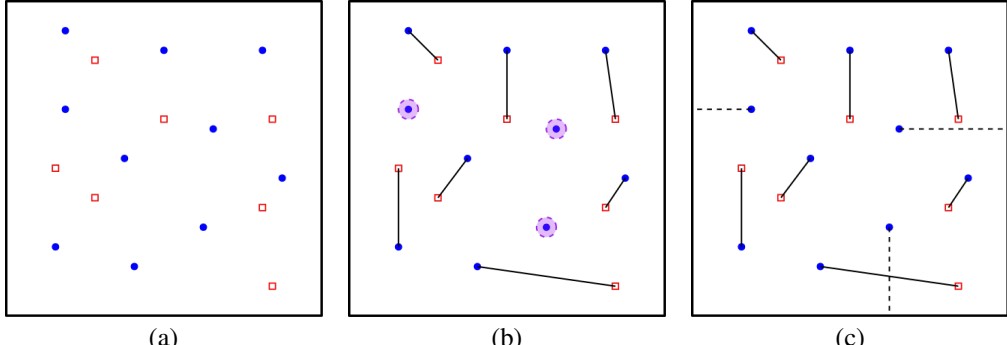

Figure 2: (a) A square $\square$ with 7 points of $A_\square$ (red squares) and 10 points of $B_\square$ (blue disks), (b) we remove $\mathrm{exc}(\square) = 3$ points selected uniformly at random from $B_\square$ (the points highlighted by purple circles) and compute the minimum-cost matching $M$ on the remaining points (solid line segments), (c) the $\square$-constrained cost of $M$ is computed as the total cost of the matching $M$ and the distance of the removed $\mathrm{exc}(\square)$ points to the boundary of $\square$ (dashed line segments).

$n_a = |A_\square|$ and $n_b = |B_\square|$. Let $m = \min\{n_a, n_b\}$ and let the *excess* of $\square$, denoted by $\mathrm{exc}(\square)$, be the difference in the number of points of $B_\square$ and $A_\square$, i.e., $\mathrm{exc}(\square) = |n_a - n_b|$. Figure 2(a) for instance shows an example of a square $\square$ with $n_a = 7, n_b = 10, m = 7$, and $\mathrm{exc}(\square) = 3$.

We prove the claimed upperbound on the expected $\square$-constrained cost of $M_\square^*$ by computing a matching $M$ whose expected $\square$-constrained cost is $O(\ell_\square \sqrt{n} \log n)$ as follows. We first remove $\mathrm{exc}(\square)$ points selected uniformly at random from $A_\square$ (resp. $B_\square$) given $n_a \geq n_b$ (resp. $n_a < n_b$) and obtain two point sets $A'$ and $B'$, both of size $m$. Let $M$ be a perfect matching from $B'$ to $A'$. In the example of Figure 2(b), three points are selected uniformly at random from $B_\square$ (the points highlighted by the purple circles) and a minimum-cost matching is computed on the remaining points. Note that the $\square$-constrained cost of $M$ is an upperbound on the $\square$-constrained cost of $M_\square^*$. Since each removed point has a distance of at most $\frac{\ell_\square}{2}$ to the boundary of $\square$,

$$w_\square(M_\square^*) \leq w_\square(M) \leq w(M) + \frac{\ell_\square}{2}\mathrm{exc}(\square). \tag{4}$$

Figure 2(c) shows the $\square$-constrained cost of the matching $M$. To bound the expected $\square$-constrained cost of $M$, first in Lemma A.3 below, we show that $\mathbb{E}\left[\mathrm{exc}(\square)\right] = O(\sqrt{n \log n})$. We then show that $\mathbb{E}\left[w(M)\right] = O(\ell_\square \sqrt{n} \log n)$. By plugging these bounds to the RHS of Equation (4), we then conclude that $\mathbb{E}\left[w_\square(M_\square^*)\right] = O(\ell_\square \sqrt{n} \log n)$, as desired.

**Lemma A.3.** *For any square $\square$ of $\mathcal{Q}$, $\mathbb{E}\left[\mathrm{exc}(\square)\right] = O(\sqrt{n \log n})$.*

*Proof.* Let $p_\square$ denote the probability that a point drawn from $\mu$ lies inside $\square$. Using the Chernoff's bound,

$$\Pr\left[|B_\square| \geq np_\square + c\sqrt{np_\square \log n}\right] \leq \frac{1}{2n^2},$$

for some constant $c \leq 3$. Similarly,

$$\Pr\left[|A_\square| \leq np_\square - c\sqrt{np_\square \log n}\right] \leq \frac{1}{2n^2}.$$

Therefore, with probability at least $1 - \frac{1}{n^2}$,

$$|B_\square| - |A_\square| \leq 2c\sqrt{np_\square \log n} = O(\sqrt{n \log n}).$$

Similarly, we can show that with probability at least $1 - \frac{1}{n^2}$, $|A_\square| - |B_\square| = O(\sqrt{n \log n})$. Since $\mathrm{exc}(\square)$ is at most $n$, we conclude that $\mathbb{E}\left[\mathrm{exc}(\square)\right] = O(\sqrt{n \log n})$. $\qquad\square$

Next, we bound $w(M)$. Since the removed points are selected uniformly at random, the two point sets $A'$ and $B'$ are two sets of $m$ i.i.d samples from the same distribution inside $\square$; in this case, the expected

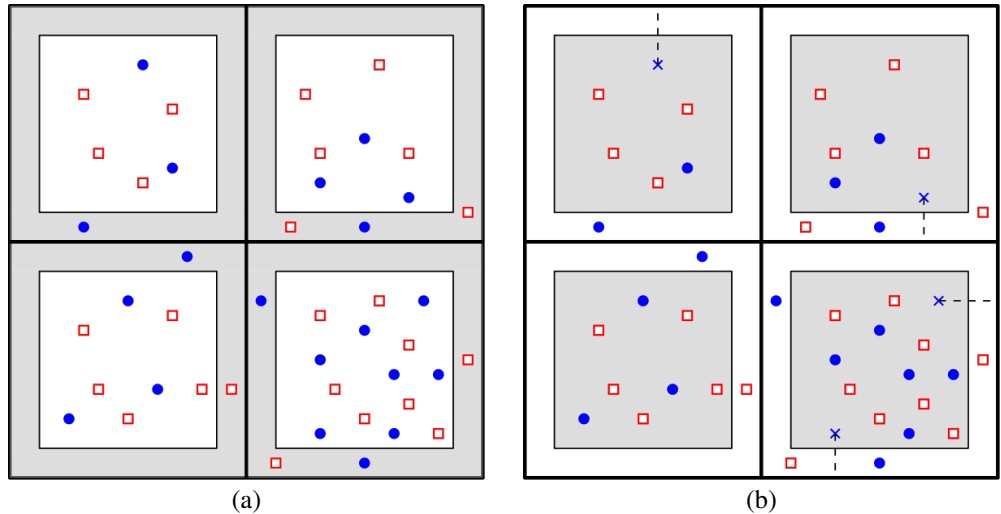

Figure 3: (a) The free points of $B_\square$ that lie inside the gray area are categorized as close, and (b) the free points of $B_\square$ (blue X marks) that lie inside the gray area are categorized as far. Each free point in this category contributes a cost of at least $n^{-1/4}\ell_\square$ to the $\square$-constrained cost of $M_\square^*$ (shown by dashed lines).

cost of the Euclidean bipartite matching (with $p = 1$) between $A'$ and $B'$ is $O(\ell_\square \sqrt{m} \log m)$ [1, 2]. Therefore,

$$\mathbb{E}\left[w(M)\right] = O(\ell_\square \sqrt{m} \log m) = O(\ell_\square \sqrt{n} \log n). \tag{5}$$

By invoking Lemma A.3 and plugging Equation (5) into Equation (4),

$$\mathbb{E}\left[w_\square(M_\square^*)\right] \le \mathbb{E}\left[w(M)\right] + \frac{\ell_\square}{2}\mathbb{E}\left[\text{exc}(\square)\right] = O(\ell_\square \sqrt{n} \log n).$$

For any dimension $d \ge 3$, $\mathbb{E}\left[w(M)\right] = \tilde{O}(\ell_\square m^{1-\frac{1}{d}}) = \tilde{O}(\ell_\square n^{1-\frac{1}{d}})$ [1, 2]. Plugging into Equation (4), we have $\mathbb{E}\left[w_\square(M_\square^*)\right] = \tilde{O}(\ell_\square n^{1-\frac{1}{d}})$, leading to the following lemma.

**Lemma A.4.** *For any square $\square$ of $\mathcal{Q}$, let $M_\square^*$ denote a $\square$-MCM on $A_\square \cup B_\square$ and let $\ell_\square$ denote the side-length of $\square$. Then, $\mathbb{E}\left[w_\square(M_\square^*)\right] = O(\ell_\square \sqrt{n} \log n)$ for 2 dimensions and $\mathbb{E}\left[w_\square(M_\square^*)\right] = \tilde{O}(\ell_\square n^{1-\frac{1}{d}})$ for $d$ dimensions, where $d \ge 3$.*

**Expected Number of Free Points.** Given any square $\square$ of $\mathcal{Q}$, using Lemma A.4, in this part we show that the expected number of free points of $B_\square$ with respect to any $\square$-MCM $M_\square^*$ is $\tilde{O}(n^{3/4})$.

We categorize any free point of $B_\square$ with respect to $M_\square^*$ as a close (resp. far) point if its distance to $\square$ is at most (resp. at least) $n^{-1/4}\ell_\square$. Let $X$ (resp. $Y$) be the set of free points of $B_\square$ that are close (resp. far). To prove Lemma 2.2, we bound the number of points in $X$ and $Y$ separately.

Recall that $n_\square^\delta$ denotes the number of points that are $\delta$-close to $\square$. Since each point in $X$ is $(n^{-1/4})$-close to $\square$, $|X| \le n_\square^\delta$ for $\delta = n^{-1/4}$ (see Figure 3(a)). Therefore, by Lemma 2.1,

$$\mathbb{E}\left[|X|\right] \le \mathbb{E}\left[n_\square^\delta\right] = O(n^{-1/4}\mathbb{E}\left[n_\square\right]) = O(n^{3/4}).$$

Furthermore, for any free point $b \in Y$, by definition, $\mathrm{d}(b, \square) > n^{-1/4}\ell_\square$. Since $b$ is a free point with respect to $M_\square^*$, it contributes a cost of at least $n^{-1/4}\ell_\square$ to $w_\square(M_\square^*)$ (see Figure 3(b)). Therefore,

$$|Y| \le \frac{w_\square(M_\square^*)}{n^{-1/4}\ell_\square}.$$

By invoking Lemma A.4 on $\square$,

$$\mathbb{E}\left[|Y|\right] \le \frac{\mathbb{E}\left[w_\square(M_\square^*)\right]}{n^{-1/4}\ell_\square} = O(n^{3/4} \log n).$$

Combining the two bounds,

$$\mathbb{E}\left[\left|B_\square^F\right|\right] = \mathbb{E}\left[|X|\right] + \mathbb{E}\left[|Y|\right] = O(n^{3/4}\log n).$$

For any dimension $d \geq 3$, one can categorize any free points in $B_\square^F$ as close (resp. far) if its distance from $\square$ is at most (resp. at least) $n^{-1/2d}\ell_\square$ and easily confirm that

$$\mathbb{E}\left[\left|B_\square^F\right|\right] = \mathbb{E}\left[|X|\right] + \mathbb{E}\left[|Y|\right] = \tilde{O}(n^{1-\frac{1}{2d}}).$$

See Section B for a detailed discussion on analyzing the expected number of free points with respect to any $\square$-MCM for any $p \in [1, \infty)$ and $d \geq 2$.

**Lemma A.5.** *For any square $\square$ of the quadtree, any minimum-cost $\square$-constrained matching $M_\square^*$, and any $\delta \in (0, 1/2)$, the expected number of free points of $B_\square$ with respect to $M_\square^*$ is $\tilde{O}\left(n^{3/4}\right)$ in 2-dimensions (and $\tilde{O}(n^{1-\frac{1}{2d}})$ in d dimensions).*

Finally, combining Lemmas A.2 and A.5 gives us the following lemma.

**Lemma 2.2.** *For any square $\square$ of a randomly shifted quadtree and any minimum-cost $\square$-constrained matching $M_\square^*$, (i) the expected number of free points of $B_\square$ with respect to $M_\square^*$ is $\tilde{O}\left(n^{3/4}\right)$ in 2-dimensions (and $\tilde{O}(n^{1-\frac{1}{2d}})$ in d dimensions), and (ii) if $\square$ is the root square, then $M_\square^*$ is a minimum-cost perfect matching on $A \cup B$.*

### A.3 Properties of the Constrained Feasibility

In this section, we first show that for any square $\square$ of $\mathcal{Q}$, any $\square$-optimal matching is a $\square$-MCM, hence proving Lemma 2.3. We then show that the combination of $\square'$-optimal matching for all children $\square'$ of $\square$ results in a $\square$-feasible matching, leading to Lemma 2.4.

For a square $\square$ of $\mathcal{Q}$, let $M, y(\cdot)$ be any $\square$-optimal matching and $M_\square^*$ be any $\square$-MCM on $A_\square \cup B_\square$. By $\square$-feasibility conditions and since $M$ is a $\square$-optimal matching,

$$w_\square(M) = \sum_{(a,b)\in M} \|a-b\| + \sum_{b\in B_M^F} \mathrm{d}(b,\square) = \sum_{(a,b)\in M}(y(b)-y(a)) + \sum_{b\in B_M^F} y(b)$$

$$= \sum_{b\in B_\square} y(b) - \sum_{a\in A_\square} y(a),$$

where the second equality comes from feasibility conditions (5) and (6), and the last equality holds due to feasibility condition (7). Similarly,

$$w_\square(M_\square^*) = \sum_{(a,b)\in M_\square^*} \|a-b\| + \sum_{b\in B_{M_\square^*}^F} \mathrm{d}(b,\square) \geq \sum_{(a,b)\in M_\square^*}(y(b)-y(a)) + \sum_{b\in B_{M_\square^*}^F} y(b)$$

$$\geq \sum_{b\in B_\square} y(b) - \sum_{a\in A_\square} y(a).$$

Combining the two bounds, $w_\square(M_\square^*) \geq \sum_{b\in B_\square} y(b) - \sum_{a\in A_\square} y(a) = w_\square(M)$. However, note that $M_\square^*$ is a $\square$-MCM; hence, we have $w_\square(M_\square^*) = w_\square(M)$ and $M$ is also a $\square$-MCM, leading to the following lemma.

**Lemma 2.3.** *Let $M_\square, y(\cdot)$ be a $\square$-optimal matching on $A_\square \cup B_\square$. Then, $M_\square$ is a minimum-cost $\square$-constrained matching.*

Next, suppose $\square_i$, $i \in [1,4]$ is the set of all children of $\square$ and let $M_i, y(\cdot)$ denote a $\square_i$-optimal matching for any $i \in [1,4]$. Define $M = \bigcup_{i=1}^4 M_i$ to be the union of the four matchings computed at the children of $\square$. In this part, we show that $M$ is a $\square$-feasible matching. To do so, we show that the matching $M$ along with the dual weights $y(\cdot)$ satisfy the constrained feasibility conditions (4) –(7).

For any pair of points $(a, b) \in A \times B$,

- if $a$ and $b$ lie inside the same child $\square'$ of $\square$, by $\square'$-feasibility conditions, conditions (4) and (5) are satisfied;

- otherwise, $a$ and $b$ lie inside different children $\square_a$ and $\square_b$ of $\square$. In this case, by construction, $(a, b)$ is a non-matching edge. By definition, $y(a) \geq 0$. Furthermore, by $\square_b$-feasibility condition (6), $y(b) \leq d(b, \square_b)$. Since $a \notin \square_b$, $\|a - b\| \geq d(b, \square_b)$ and as a result,
$$y(b) - y(a) \leq y(b) \leq d(b, \square_b) \leq \|a - b\|.$$
Therefore, conditions (4) and (5) are satisfied in this case as well.

For any point $b \in B$, let $\square_i$ be the child of $\square$ containing $b$. By $\square_i$-feasibility conditions,
$$y(b) \leq d(b, \square_i) \leq d(b, \square).$$
Furthermore, for any free point $a \in A^F$, suppose $a$ lies inside a child $\square_i$ of $\square$. Since $a$ is also a free point in $M_i$, by the $\square_i$-feasibility conditions, $y(a) = 0$; thus, feasibility conditions (6) and (7) hold as well and $M, y(\cdot)$ is a $\square$-feasible matching.

**Lemma 2.4.** *For any square $\square$, let $\square_i$, $i \in [1, 4]$ be the set of all children of $\square$ and let $M_i, y(\cdot)$ denote a $\square_i$-optimal matching. Then, the matching $\bigcup_{i=1}^{4} M_i, y(\cdot)$ is a $\square$-feasible matching.*

Note that the discussion above is not dependent on the dimension and is also applicable to the $d$-dimensional space for any $d \geq 2$.

**Corollary A.6.** *For any $d$-dimensional hypercube $\square$, let $\mathsf{C}[\square]$ be the set of all children of $\square$ and let $M_{\square'}, y(\cdot)$ denote a $\square'$-optimal matching for any child $\square' \in \mathsf{C}[\square]$. Then, the matching $\bigcup_{\square' \in \mathsf{C}[\square]} M_{\square'}, y(\cdot)$ is a $\square$-feasible matching.*

## A.4 Properties of the AUGMENT procedure

Given a square $\square$ of $\mathcal{Q}$, a $\square$-feasible matching $M_\square, y(\cdot)$, and an admissible path $P$, in this section we show that augmenting $M_\square$ along $P$ does not violate the feasibility conditions (4)–(7). Furthermore, the augmentation reduces the number of $\square$-free points in $\mathcal{B}_\square^F$ by one. The combination of these properties proves Lemma 2.5.

Note that any free point $b \in B$ (resp. $a \in A$) after augmentation was also a free point prior to augmentation. Therefore, the feasibility condition (6) (resp. (7)) holds for $b$ (resp. $a$) since the augmentation does not change the dual weights and the same condition holds prior to augmentation. In other words, all dual weights will remain satisfying the feasibility conditions (6) and (7) after augmentation. For all edges $(a, b) \in A \times B$ such that $(a, b) \notin P$, if $(a, b)$ is a matching (resp. non-matching) edge prior to augmentation, it remains a matching (resp. non-matching) edge after augmentation and the condition (5) (resp. (4)) for $(a, b)$ remains satisfied. For any edge $(a, b)$ in the admissible path $P$, due to the admissibility of the edge, $y(b) - y(a) = \|a - b\|$. Two cases can happen.

- If $(a, b) \notin M_\square$ prior to augmentation, it will be a matching edge after the augmentation and condition (5) holds for $(a, b)$.
- If $(a, b) \in M_\square$ prior to augmentation, it will be a non-matching edge after the augmentation and condition (4) holds for $(a, b)$.

Hence, the matching $M_\square, y(\cdot)$ remain $\square$-feasible after augmentation. Next, we show that the number of $\square$-free points of $B_\square$ reduces by one after augmenting $M_\square$ along $P$. Recall that the admissible path $P$ is a path from a $\square$-free point $b \in \mathcal{B}_\square^F$ to either a free point $a \in A_\square^F$ or a zero-slack point $b' \in B_\square$. In both cases, the point $b$ becomes a matched point after the augmentation and will be removed from the set of $\square$-free points $\mathcal{B}_\square^F$. In the second case, although after the augmentation the point $b'$ becomes an unmatched point, since it has a zero slack, $b'$ will not be a $\square$-free point. Finally, other than the two endpoints of $P$, all other points of $P$ are matched points both before and after the augmentation. Thus, the size of $\mathcal{B}_\square^F$ reduces by one.

**Lemma 2.5.** *Suppose $M_\square, y(\cdot)$ is a $\square$-feasible matching and $P$ is an admissible path. After augmenting $M_\square$ along $P$, the matching $M_\square, y(\cdot)$ remains $\square$-feasible. Furthermore, the augmentation reduces the number of $\square$-free points of $B_\square$ with respect to $M_\square$ by one.*

## A.5 Properties of the CONSTRAINEDHUNGARIANSEARCH procedure

Given a square $\square$ of the quadtree and a $\square$-feasible matching $M_\square, y(\cdot)$, in this section we show that after executing the CONSTRAINEDHUNGARIANSEARCH procedure for $\square$, the updated dual weights will be $\square$-feasible and the returned path $P$ will be an admissible path, resulting in Lemma 3.1.

Let $y(\cdot)$ (resp. $y'(\cdot)$) denote the dual weights of the points before (resp. after) the execution of the CONSTRAINEDHUNGARIANSEARCH procedure. For any free point $a \in A_\square^F$, by the definition, $\kappa_a \geq \kappa_v$. Therefore, the CONSTRAINEDHUNGARIANSEARCH procedure does not update the dual weight of $a$ and $y'(a) = y(a)$, which by $\square$-feasibility conditions on $M_\square, y(\cdot)$ is zero; thus, the updated dual weights on free point in $A_\square^F$ meet condition (7).

For any point $b \in B_\square$, if $\kappa_b \geq \kappa$, then the procedure will not update the dual weight of $b$ and condition (6) remains satisfied for $b$. Otherwise, $\kappa_b < \kappa$. By the definition, $\kappa \leq \kappa_b + s_\square(b)$, where $s_\square(b)$ denotes the slack of $b$ prior to dual updates. Therefore,

$$y'(b) = y(b) - \kappa_b + \kappa \leq y(b) + s_\square(b) = \mathrm{d}(\square, b).$$

As a result, the dual updates do not violate the feasibility condition (6). Furthermore, if $\kappa = \kappa_b + s_\square(b)$, then $y'(b) = \mathrm{d}(b, \square)$ and $b$ will be a zero-slack point.

For any edge $(u, v)$, let $s(u, v)$ denote the slack of $(u, v)$ before the dual updates. For any matching edge $(a, b)$, since the only incoming edge to $b$ in the residual network is the zero-weight edge $(a, b)$, $\kappa_b = \kappa_a$. Thus,

$$y'(b) - y'(a) = (y(b) - \kappa_b + \kappa) - (y(a) - \kappa_a + \kappa) = y(b) - y(a) = \|a - b\|.$$

Similarly, for any non-matching edge $(b, a)$, $\kappa_a \leq \kappa_b + s(b, a)$, since there is a direct edge from $b$ to $a$ in the residual network with a weight $s(b, a)$. Hence,

$$y'(b) - y'(a) = (y(b) - \kappa_b + \kappa) - (y(a) - \kappa_a + \kappa) \leq (y(b) - y(a)) + s(b, a) = \|a - b\|.$$

Therefore, the updated dual weights will remain $\square$-feasible. Finally, if $(b, a)$ is an edge on the shortest path tree constructed by the Dijkstra's algorithm, then $\kappa_a = \kappa_b + s(b, a)$ and therefore, $y'(b) - y'(a) = \|a - b\|$, i.e., $(b, a)$ is an admissible edge after the dual updates. Since all edges of $P$ are the edges of the shortest path tree, all such edges are admissible after the dual updates. Furthermore, if $P$ ends at a vertex $u \in B_\square$, then by the discussion above, $u$ will be a zero-slack point. Therefore, $P$ is an admissible path.

**Lemma 3.1.** *Given any square $\square$ of $Q$ and any $\square$-feasible matching $M_\square, y(\cdot)$, after executing the* CONSTRAINEDHUNGARIANSEARCH *procedure on $\square$, the updated dual weights remain $\square$-feasible. Furthermore, the returned path $P$ is an admissible path.*

## B  Our Results in General Settings

Given two sets of $n$ i.i.d samples $A$ and $B$ from a distribution $\mu$ inside the unit $d$-dimensional hypercube and any integer $p \in [1, \infty)$, in this section, we analyze the execution time of our algorithm for computing the $p$th power Euclidean bipartite matching from $B$ to $A$.

Recall that $\mathsf{C}[\square]$ denotes the set of non-empty children of $\square$. In the conquer step of our algorithm on $\square$, our algorithm constructs a matching $M_\square$ as $M_\square = \bigcup_{\square' \in \mathsf{C}[\square]} M_{\square'}$. To analyze the running time, similar to Section 3.2, we bound the number of iterations by the number of free points with respect to $M_\square$. Similar to Section A.2, for any child $\square' \in \mathsf{C}[\square]$, we categorize any free points in $B_{\square'}^F$ with respect to $M_{\square'}$ as close (resp. far) if its distance to the boundary of $\square'$ is at most (resp. at least) $n^{-\alpha} \ell_\square$, where

$$\alpha = \begin{cases} \frac{p}{(p+1)d}, & p < \frac{d}{2}, \\ \frac{1}{2(p+1)}, & p \geq \frac{d}{2}, \end{cases}$$

and define the set $X_{\square'}$ (resp $Y_{\square'}$) as the subset of free points of $B_{\square'}^F$ that are close (resp. far). In this way, we can express the number of free points of $B_\square^F$ as

$$\left| B_\square^F \right| = \sum_{\square' \in \mathsf{C}[\square]} \left| B_{\square'}^F \right| = \sum_{\square' \in \mathsf{C}[\square]} |X_{\square'}| + \sum_{\square' \in \mathsf{C}[\square]} |Y_{\square'}|. \tag{6}$$

From Lemma A.1, for any child $\square' \in \mathsf{C}[\square]$ and any $\delta \in (0, 1/2)$, $\mathbb{E}\left[n_{\square'}^\delta\right] = O(d\delta \mathbb{E}[n_{\square'}])$. By setting $\delta = n^{-\alpha}$ and using $\sum_{\square' \in \mathsf{C}[\square]} n_{\square'} \leq n$,

$$\sum_{\square' \in \mathsf{C}[\square]} \mathbb{E}[|X_{\square'}|] = \begin{cases} O\left(dn^{-\frac{p}{(p+1)d}} \mathbb{E}\left[\sum_{\square' \in \mathsf{C}[\square]} n_{\square'}\right]\right), p < \frac{d}{2}, \\ O\left(dn^{-\frac{1}{2(p+1)}} \mathbb{E}\left[\sum_{\square' \in \mathsf{C}[\square]} n_{\square'}\right]\right), p \geq \frac{d}{2}, \end{cases} = \begin{cases} O\left(dn^{1-\frac{p}{(p+1)d}}\right), p < \frac{d}{2}, \\ O\left(dn^{1-\frac{1}{2(p+1)}}\right), p \geq \frac{d}{2}. \end{cases}$$

Next, we bound the second term in the RHS of Equation (6). By Corollary A.6, the matching $M_\square$ is a $\square$-feasible matching. Let $M_\square^*$ be any $\square$-MCM. In the following lemma, we show that the total constrained costs of the matchings computed at the children of $\square$ is upper-bounded by the $\square$-constrained cost of $M_\square^*$.

**Lemma B.1.** *Given any square $\square$ of the quadtree, let $M_\square^*$ be any $\square$-MCM and for any child $\square' \in \mathsf{C}[\square]$, let $M_{\square'}, y(\cdot)$ be a $\square'$-optimal matching for $\square'$. Then, $\sum_{\square' \in \mathsf{C}[\square]} w_{\square'}(M_{\square'}) \le w_\square(M_\square^*)$.*

*Proof.* By $\square$-feasibility conditions,

$$w_\square(M_\square^*) = \sum_{(a,b) \in M_\square^*} \|a - b\|^p + \sum_{b \in B_{M_\square^*}^F} \mathrm{d}(b, \square)^p \ge \sum_{(a,b) \in M_\square^*} (y(b) - y(a)) + \sum_{b \in B_{M_\square^*}^F} y(b)$$

$$\ge \sum_{b \in B_\square} y(b) - \sum_{a \in A_\square} y(a).$$

Furthermore, for any child $\square'$ of $\square$,

$$w_{\square'}(M_{\square'}) = \sum_{(a,b) \in M_{\square'}} \|a - b\|^p + \sum_{b \in B_{M_{\square'}}^F} \mathrm{d}(b, \square')^p = \sum_{(a,b) \in M_{\square'}} (y(b) - y(a)) + \sum_{b \in B_{M_{\square'}}^F} y(b)$$

$$= \sum_{b \in B_{\square'}} y(b) - \sum_{a \in A_{\square'}} y(a).$$

Summing over all children of $\square$,

$$\sum_{\square' \in \mathsf{C}[\square]} w_{\square'}(M_{\square'}) = \sum_{\square' \in \mathsf{C}[\square]} \left( \sum_{b \in B_{\square'}} y(b) - \sum_{a \in A_{\square'}} y(a) \right) = \sum_{b \in B_\square} y(b) - \sum_{a \in A_\square} y(a) \le w_\square(M_\square^*).$$

$\square$

From Lemma B.1 and since the contribution of each free point in $Y$ is at least $(\ell_\square n^{-\alpha})^p$,

$$\sum_{\square' \in \mathsf{C}[\square]} \mathbb{E}\left[|Y_{\square'}|\right] \le \sum_{\square' \in \mathsf{C}[\square]} \frac{\mathbb{E}\left[w_{\square'}(M_{\square'}^*)\right]}{(n^{-\alpha}\ell_\square)^p} \le \frac{\mathbb{E}\left[w_\square(M_\square^*)\right]}{(n^{-\alpha}\ell_\square)^p}. \tag{7}$$

Combining the results in [1, 2] and Lemma A.3,

$$\mathbb{E}\left[w_\square(M_\square^*)\right] = \begin{cases} \tilde{O}(\ell_\square^p n^{1-\frac{p}{d}}), & p < \frac{d}{2}, \\ \tilde{O}(\ell_\square^p n^{\frac{1}{2}}), & p \ge \frac{d}{2}, \end{cases}$$

Plugging in Equation (7),

$$\mathbb{E}\left[|Y|\right] = \sum_{\square' \in \mathsf{C}[\square]} \mathbb{E}\left[|Y_{\square'}|\right] = \begin{cases} \frac{\mathbb{E}\left[w(M_\square^*)\right]}{\left(\ell_\square n^{-\frac{p}{(p+1)d}}\right)^p} = \tilde{O}(n^{1-\frac{p}{d}+\frac{p^2}{(p+1)d}}) = \tilde{O}(n^{1-\frac{p}{(p+1)d}}), & p < \frac{d}{2}, \\ \frac{\mathbb{E}\left[w(M_\square^*)\right]}{\left(\ell_\square n^{-\frac{1}{2(p+1)}}\right)^p} = \tilde{O}(n^{\frac{1}{2}+\frac{p}{2(p+1)}}) = \tilde{O}(n^{1-\frac{1}{2(p+1)}}), & p \ge \frac{d}{2}. \end{cases}$$

Therefore, the total number of iterations of the conquer step of our algorithm on any hypercube $\square$ is, in expectation, bounded by

$$\mathbb{E}\left[|B_\square^F|\right] = \sum_{\square' \in \mathsf{C}[\square]} \mathbb{E}\left[|X_{\square'}|\right] + \sum_{\square' \in \mathsf{C}[\square]} \mathbb{E}\left[|Y_{\square'}|\right] = \begin{cases} \tilde{O}(n^{1-\frac{p}{(p+1)d}}), & p < \frac{d}{2}, \\ \tilde{O}(n^{1-\frac{1}{2(p+1)}}), & p \ge \frac{d}{2}. \end{cases}$$

Each iteration takes $\tilde{O}(n_\square \Phi(n_\square))$ time. Recall that $T_i$ denotes the total total execution time of the conquer step of our algorithm across all hypercubes of any level $i$ of the quadtree and $\mathcal{L}(i)$ denotes the set of all hypercubes of the quadtree at level $i$.

$$\mathbb{E}\left[T_i\right] = \begin{cases} \sum_{\square \in \mathcal{L}(i)} \tilde{O}(n^{1-\frac{p}{(p+1)d}} n_\square \Phi(n_\square)), & p < \frac{d}{2}, \\ \sum_{\square \in \mathcal{L}(i)} \tilde{O}(n^{1-\frac{1}{2(p+1)}} n_\square \Phi(n_\square)), & p \ge \frac{d}{2}, \end{cases} = \begin{cases} \tilde{O}(n^{2-\frac{p}{(p+1)d}} \Phi(n)), & p < \frac{d}{2}, \\ \tilde{O}(n^{2-\frac{1}{2(p+1)}} \Phi(n)), & p \ge \frac{d}{2}. \end{cases}$$

Summing over all $O(\log \Delta)$ levels of the quadtree, we conclude the following theorem.

**Theorem B.2.** *There exists a randomized algorithm that, given any two point sets $A$ and $B$ sampled independently and identically from a distribution $\mu$ inside the unit $d$-dimensional hypercube, where $|A| = |B| = n$ and $\mu$ is not known to the algorithm, and a parameter $p \in [1, \infty)$, computes an exact $p$th power Euclidean bipartite matching between $A$ and $B$ and has an expected running time of*

$$\begin{cases} \tilde{O}(n^{2 - \frac{p}{(p+1)d}} \Phi(n) \log \Delta), & p < \frac{d}{2}, \\ \tilde{O}(n^{2 - \frac{1}{2(p+1)}} \Phi(n) \log \Delta), & p \geq \frac{d}{2}. \end{cases}$$

*Here, $\Delta$ is the spread of the points in $A \cup B$.*

Suppose the distribution $\mu$ inside the unit $d$-dimensional hypercube satisfies the following property.

(P) There exists a constant $\tau \geq 1$ such that for any $d$-dimensional ball $\mathcal{B}$ of radius $n^{-\tau}$,

$$\int_R \mu(x)dx \leq \frac{1}{2n^3}.$$

Let $A$ and $B$ be two sets of $n$ i.i.d samples from the distribution $\mu$. For any point $u \in A \cup B$, let $\mathcal{B}(u, r)$ denote a ball of radius $r$ centered at the point $u$. From property (P),

$$\Pr\left[\left|(A \cup B) \cap \mathcal{B}(u, n^{-\tau})\right| > 1\right] \leq \sum_{v \in (A \cup B) \setminus \{u\}} \Pr\left[v \in \mathcal{B}(u, n^{-\tau})\right] \leq \frac{1}{2n^2}.$$

Let $C_{\min}$ (resp. $C_{\max}$) denote the minimum (resp. maximum) pairwise Euclidean distance of the points in $A \cup B$, i.e., $C_{\min} := \min_{u,v \in A \cup B, u \neq v} \|u - v\|$ and $C_{\max} := \max_{u,v \in A \cup B, u \neq v} \|u - v\|$. Then,

$$\Pr\left[C_{\min} \leq n^{-\tau}\right] \leq \sum_{u \in (A \cup B)} \Pr\left[\left|(A \cup B) \cap \mathcal{B}(u, n^{-\tau})\right| > 1\right] \leq \frac{1}{n}.$$

Therefore, $\Pr\left[C_{\min} > n^{-\tau}\right] = 1 - \Pr\left[C_{\min} \leq n^{-\tau}\right] \geq 1 - \frac{1}{n}$. Note that one can always shift and scale up the coordinates of the points inside the unit hypercube so that $C_{\max} \geq 1$. Shifting and scaling the coordinates does not change the final matching. Also, since all points are inside the unit hypercube, $C_{\max} \leq \sqrt{d}$. Therefore, with high probability, the spread $\Delta$ of the points in $A \cup B$ is bounded by

$$\Delta = \frac{C_{\max}}{C_{\min}} \leq \sqrt{d} n^{\tau},$$

and as a result, the height of the quadtree, with a high probability, would be $O(\log \Delta) = O(\tau \log nd)$, resulting in the following corollary.

**Corollary B.3.** *There exists a randomized algorithm that, given any two point sets $A$ and $B$ sampled independently and identically from a distribution $\mu$ inside the unit $d$-dimensional hypercube, where $|A| = |B| = n$, $\mu$ is not known to the algorithm and satisfies the condition of property (P), and a parameter $p \in [1, \infty)$, computes an exact $p$th power Euclidean bipartite matching between $A$ and $B$ and has an expected running time of*

$$\begin{cases} \tilde{O}(n^{2 - \frac{p}{(p+1)d}} \Phi(n)), & p < \frac{d}{2}, \\ \tilde{O}(n^{2 - \frac{1}{2(p+1)}} \Phi(n)), & p \geq \frac{d}{2}. \end{cases}$$

## C  Additional Experiments

In this section, we present the results of our additional experiments. In these experiments, we use two additional datasets: (1) Exponential distribution in the plane (Exponential), and (2) a Gaussian mixture model consisting of 10 clusters in the 2-dimensional space (Clustered), where the centers of the clusters are chosen uniformly at random. Since these distributions are not bounded to the unit square, after drawing samples from each one, we normalize the coordinates of the points to lie inside the unit square.

Figure 4 shows the results of our experiments comparing the running time of our algorithm with the Hungarian algorithm for computing the Euclidean bipartite matching ($p = 1$) as well as 2nd power Euclidean bipartite matching ($p = 2$, also known as the Root Mean Squared (RMS) matching). We

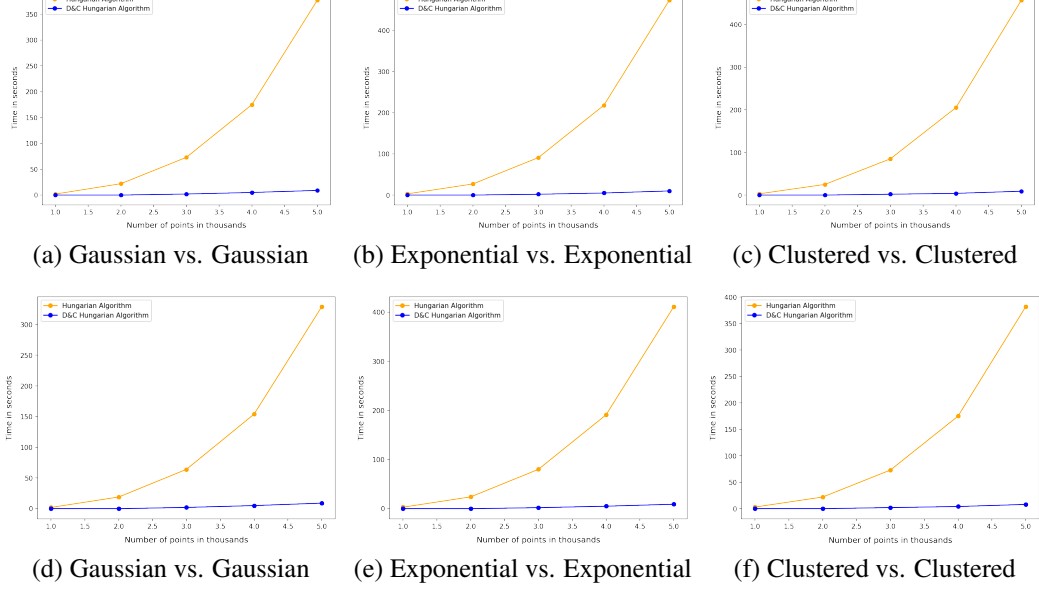

(a) Gaussian vs. Gaussian   (b) Exponential vs. Exponential   (c) Clustered vs. Clustered

(d) Gaussian vs. Gaussian   (e) Exponential vs. Exponential   (f) Clustered vs. Clustered

Figure 4: Running times of our divide-and-conquer algorithm and the Hungarian algorithm for computing the (a)–(c) Euclidean bipartite matching ($p = 1$) and (d)–(f) squared Euclidean bipartite matching ($p = 2$) between samples drawn from the same synthetic datasets.

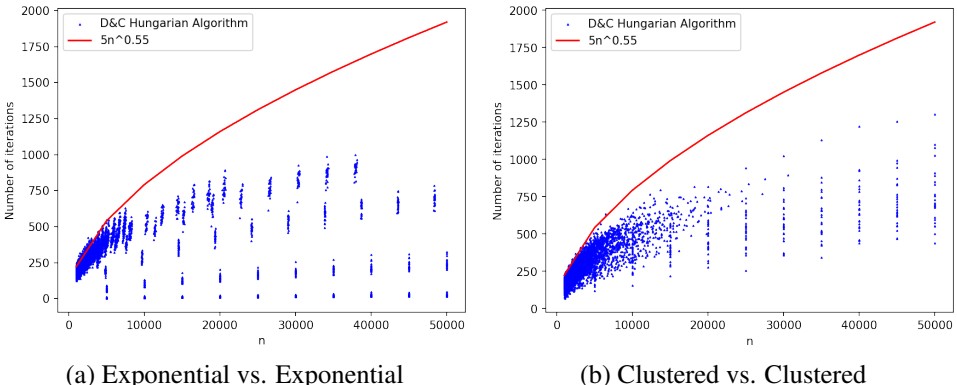

(a) Exponential vs. Exponential   (b) Clustered vs. Clustered

Figure 5: The number of iterations of the conquer step for a square with $n$ points for samples from (a) Exponential vs. Exponential, and (b) Clustered vs. Clustered distributions.

observe that in all cases, our divide-and-conquer algorithm outperforms the Hungarian algorithm and computes the exact matching significantly faster.

In Figure 5, we show the number of iterations of our algorithm on a square with $n$ points inside the square. We observe that for all datasets, the number of iterations is much lower than the upper-bound of $\tilde{O}(n^{3/4})$ we showed in our analysis.