# OpenReview forum: "A Robust Exact Algorithm for the Euclidean Bipartite Matching Problem"
_NeurIPS.cc/2023/Conference — NeurIPS 2023 poster_

### Official Review · Reviewer_L63E · 2023-06-23

**Soundness:** 3 good
**Presentation:** 3 good
**Contribution:** 2 fair
**Rating:** 5
**Confidence:** 4

**Summary:**

The paper presents a randomized algorithm for computing a minimum-cost matching of a bipartite graph induced by two point-sets, A and B, in the Euclidean plane. The best running time for the problem under consideration is n^2polylog(n). When A and B are drawn independently and identically from a fixed probability distribution, the presented algorithm achieves a running time of n^{7/4}polylog(n+D), where D is the ratio between the distance of the farthest pair of points to that of the closest pair of points. The presented algorithm can potentially improve the quasi-quadratic bound under the probability assumption and the assumption that D is not too large. The algorithm generalizes to higher dimensions.



**Strengths:**

The minimum-cost matching is an important combinatorial problem, even on bipartite geometric graphs.

**Weaknesses:**


* The improvement over existing bounds is only under certain probabilistic assumptions, and assumptions about the relative positions of the points in the plane. Moreover, the presented algorithm is randomized.
* I don’t believe that the results will appeal to the broad audience of NeurIPS. The paper does not motivate the problem well with respect to the scope of NeurIPS. The problem is very restricted.
* The techniques are not very novel; they are adaptations of the Hungarian method plus a clever use of data structures.


**Questions:**

Could you please explain the relevance of your results to NeurIPS, beyond the importance of the matching problem as a generic optimization problem?

---

> ### Author Rebuttal · Authors · 2023-08-09
>
> We would like to thank you for reviewing our submission.
>
> > Could you please explain the relevance of your results to NeurIPS, beyond the importance of the matching problem as a generic optimization problem?
>
> Minimum-cost bipartite matching is extensively used in many applications in Machine Learning, Computer Vision, and Statistical Inference. We enumerate a few of these below.
>
> **Machine learning applications:** Optimal bipartite matching and its cost is used (i) as a loss function in GANs [1, 2], (ii) within autoencoders [3, 4], (iii) in domain adaptation [5, 6], (iv) clustering [7], and (v) self-supervised learning [8].
>
> **Computer vision applications:** Multi-object tracking [9],  object-centric learning [10, 11], object detection [12, 13], image retrieval [14], instance segmentation [15, 16], and vector graphics [17].
>
> **Statistical inference applications:** Two-sample test [18, 19], mutual-independence test [20], and distributional shifts [21].
>
> Computing exact matchings is too expensive in many of these applications. This has motivated machine learning researchers to design approximation algorithms [22--28]. Our contribution is to obtain asymptotically faster exact algorithm for stochastic inputs such as those that arise in some of the applications above.
>
> ---
> ---
>
> **References.**
>
> [1] H. Liu, G. U. Xianfeng, and D. Samaras. "A two-step computation of the exact gan wasserstein distance." In ICML, 2018.
>
> [2] J. Cao, L. Mo, Y. Zhang, K. Jia, C. Shen, and M. Tan. "Multi-marginal wasserstein gan." NeurIPS, 2019.
>
> [3] A. Kosiorek, S. Sabour, Y. W. Teh, and G. E. Hinton. "Stacked capsule autoencoders." NeurIPS, 2019.
>
> [4] S. Kolouri, P. E. Pope, C. E. Martin, and G. K. Rohde. "Sliced Wasserstein auto-encoders." In ICLR, 2018.
>
> [5] Y. Balaji, R. Chellappa, and S. Feizi. "Robust optimal transport with applications in generative modeling and domain adaptation." NeurIPS, 2020.
>
>
> [6] C. Lee, T. Batra, M. Haris Baig, and D. Ulbricht. "Sliced wasserstein discrepancy for unsupervised domain adaptation." In Proceedings of the IEEE/CVF conference on computer vision and pattern recognition, 2019.
>
> [7] X. Yang, C. Deng, K. Wei, J. Yan, and W. Liu. "Adversarial learning for robust deep clustering." NeurIPS, 2020.
>
> [8] X. Wen, B. Zhao, A. Zheng, X. Zhang, and X. Qi. "Self-supervised visual representation learning with semantic grouping." NeurIPS, 2022.
>
> [9] Y. Zhang, P. Sun, Y. Jiang, D. Yu, F. Weng, Z. Yuan, P. Luo, W. Liu, and X. Wang. "Bytetrack: Multi-object tracking by associating every detection box." In ECCV, 2022.
>
> [10] F. Locatello, D. Weissenborn, T. Unterthiner, A. Mahendran, G. Heigold, J. Uszkoreit, A. Dosovitskiy, and T. Kipf. "Object-centric learning with slot attention." NeurIPS, 2020.
>
> [11] J. Brady, R. S. Zimmermann, Y. Sharma, B. Schölkopf, J. von Kügelgen, and W. Brendel. "Provably Learning Object-Centric Representations." arXiv preprint, 2023.
>
> [12] Y. Wang, and J. M. Solomon. "Object dgcnn: 3d object detection using dynamic graphs." NeurIPS, 2021.
>
> [13] Y. Li, Y. Chen, X. Qi, Z. Li, J. Sun, and J. Jia. "Unifying voxel-based representation with transformer for 3d object detection." NeurIPS, 2022.
>
> [14] Y. Rubner, C. Tomasi, and L. J. Guibas. "The earth mover's distance as a metric for image retrieval." International journal of computer vision, 2000.
>
> [15] B. Dong, F. Zeng, T. Wang, X. Zhang, and Y. Wei. "Solq: Segmenting objects by learning queries." NeurIPS, 2021.
>
> [16] S. Hwang, M. Heo, S. W. Oh, and S. J. Kim. "Video instance segmentation using inter-frame communication transformers." NeurIPS, 2021.
>
> [17] A. Carlier, M. Danelljan, A. Alahi, and R. Timofte. "Deepsvg: A hierarchical generative network for vector graphics animation." NeurIPS, 2020.
>
> [18] M. Imaizumi, H. Ota, and T. Hamaguchi. "Hypothesis Test and Confidence Analysis With Wasserstein Distance on General Dimension." Neural Computation, 2022.
>
> [19] N. Deb, B. B. Bhattacharya, and B. Sen. "Efficiency lower bounds for distribution-free Hotelling-type two-sample tests based on optimal transport." arXiv preprint, 2021.
>
> [20] N. Deb, and B. Sen. "Multivariate rank-based distribution-free nonparametric testing using measure transportation." Journal of the American Statistical Association, 2023.
>
> [21] S. Rabanser, S. Günnemann, and Z. Lipton. "Failing loudly: An empirical study of methods for detecting dataset shift." NeurIPS, 2019.
>
>
> [22] M. Cuturi. "Sinkhorn distances: Lightspeed computation of optimal transport." NeurIPS, 2013.
>
> [23] J. Altschuler, J. Niles-Weed, and P. Rigollet. "Near-linear time approximation algorithms for optimal transport via Sinkhorn iteration." NeurIPS, 2017.
>
> [24] P. Dvurechensky, A. Gasnikov, and A. Kroshnin. "Computational optimal transport: Complexity by accelerated gradient descent is better than by Sinkhorn’s algorithm." In ICML, 2018.
>
> [25] G. Luise, A. Rudi, M. Pontil, and C. Ciliberto. "Differential properties of sinkhorn approximation for learning with wasserstein distance." NeurIPS, 2018.
>
> [26] N. Lahn, D. Mulchandani, and S. Raghvendra. "A graph theoretic additive approximation of optimal transport." NeurIPS, 2019.
>
>
> [27] J. Altschuler, F. Bach, A. Rudi, and J. Niles-Weed. "Massively scalable Sinkhorn distances via the Nyström method." NeurIPS, 2019.
>
> [28] P. K. Agarwal, S. Raghvendra, P. Shirzadian, and R. Sowle. "A Higher Precision Algorithm for Computing the $1 $-Wasserstein Distance." In ICLR, 2022.

---

> > ### Comment · Reviewer_L63E · 2023-08-12
> >
> > Thank you for the clarifications and the references. I updated my rating to "Borderline Accept".

---

### Official Review · Reviewer_SwWP · 2023-06-28

**Soundness:** 3 good
**Presentation:** 4 excellent
**Contribution:** 3 good
**Rating:** 7
**Confidence:** 3

**Summary:**

This paper studies the Euclidean bipartite matching problem. In the problem, there is a complete bipartite graph on parts A and B, and the cost of edge (alb) is ||a-b||^p. The goal is to compute the minimum cost perfect matching as quickly as possible. This setting is most motivated in the paper by computing the empirical p-Wasserstein distance (though I think the problem is also interesting in its own right). It is important to note that these edge costs are non-integral, in particular they can even be irrational. Many of the sub-quadratic algorithms that work when the edge costs are integral do not apply here.

It was known that the minimum cost matching in this setting can be computed with the Hungarian algorithm in time O(n^3), and in time O(n^2) in geometric settings. The main contribution of this work is showing that in geometric setting (i.e., the vertices of A,B are points in R^d), there exists an algorithm that runs in weakly polynomial sub-quadratic time. The “weakly polynomial” comes from a term in the runtime of log(Delta), where Delta gives the spread (i.e., ratio of the distance between the farthest and closest pair of points in A and B) of the point set, specifically the runtime is \tilde{O}(n^{2-1/(2d)}*phi(n)*log(Delta)), where the phi(n) is some runtime dealing with a nearest neighbor data structure (polylog(n) in d=2). Note that when A and B are sampled from some distribution mu in the unit square, even when mu is unknown, the runtime is \tilde{O}(n^{7/4}*log(Delta)).

The main strategy is to deploy a geometric Hungarian algorithm with a divide and conquer method using the quadtree. However, the divide portion is only helpful when points in the quad tree are not close to the boundary of their children (in some sense that I’m stating very informally). Therefore the worst-case is still just that of the Hungarian algorithm, but we get improvement when these bad boundary cases do not happen.


**Strengths:**

- Improved runtime on an interesting problem.
- Interesting technical insights in using quadtrees with a geometric Hungarian algorithm.  Even after this main idea, there are a couple hurdles the authors need to overcome.
- Techniques are relatively practical, and this is exemplified with experiments on both synthetic and real data. This is particularly interesting because a lot of the past work apparently uses complicated, hard to implement data structures.
- I thought the paper was extremely well-presented. The technical difficulties of the paper are really well laid out.

**Weaknesses:**

- It was not totally clear what techniques from the authors are totally new and what is building off of prior work. I would recommend adding a brief discussion of the work by Sharathkumar, at least. My understanding is that the big technical idea of, and then modifying it so the divide and conquer parts actually give progress, is a new perspective. Can you comment more on this?


**Questions:**

Other comments to authors:
- (see question in weaknesses)
- Great figures through out
- Nice discussion of high level technical overview at the end of Section 1
- Assuming you’re given another page in your camera ready, I’d add the conclusion section from your Appendix to the main body. The future directions/ open problems you give are nice.

---

> ### Author Rebuttal · Authors · 2023-08-09
>
> We appreciate your thorough review and feedback.
>
> > It was not totally clear what techniques from the authors are totally new and what is building off of prior work. I would recommend adding a brief discussion of the work by Sharathkumar, at least. My understanding is that the big technical idea of, and then modifying it so the divide and conquer parts actually give progress, is a new perspective. Can you comment more on this?
>
> **Response.** Your understanding of the main technical idea is correct. The novelty of our algorithm is in placing the classical Hungarian algorithm within a geometric divide-and-conquer framework.
> Our algorithm does not rely on any prior work except for a weighted nearest-neighbor based implementation of the Hungarian search step (which was proposed in [1, 2] and used in [3, 4]). Our algorithm also uses a randomly shifted quadtree, which is a popular data structure for the design of geometric approximation algorithms. To the best of our knowledge, our algorithm is the first to use them in the design of an exact Euclidean bipartite matching algorithm.
>
> **Comparison with Sharathkumar [4].** We have included a brief comparison of our result with the result of Sharathkumar [4] in lines 61--66 of our initial submission. We will extend it to also include a comparison of techniques.
>
> From a technical standpoint, apart from using a combinatorial primal-dual approach, the algorithm by Sharathkumar [4] and our algorithm are quite different.
> The algorithm in [4] uses the cost scaling framework of Gabow and Tarjan [5, 6] to find an approximate solution. Using the properties of this approximate solution, they 'trap' the edges of the optimal matching in a *planar* graph, i.e., a graph that can be drawn on a plane without any overlapping edges. They then use an algorithm by Lipton and Tarjan [7] to find an optimal matching inside this planar graph.
>
> The proof of correctness of [4] relies on (a) the $2$-dimensional geometry of the input, (b) the edge-costs being square-roots of integers (owing to integer coordinates) and (c) the triangle inequality of Euclidean distances. Thus, their algorithm does not extend to (i) higher dimensions (due to (a) not being satisfied), (ii) to points with real-valued coordinates (due to (b) not being satisfied) or (iii) to the case where edge costs are squared-Euclidean (due to (c) not being satisfied). In contrast, our algorithm extends to all these cases.
>
> ---
> ---
> **References.**
>
> [1] P. Vaidya. "Geometry helps in matching." In Proceedings of the twentieth annual ACM symposium on theory of computing, 1988.
>
> [2] R. Sharathkumar, and P. K. Agarwal. "Algorithms for the transportation problem in geometric settings." In Proceedings of the twenty-third annual ACM-SIAM symposium on Discrete Algorithms, 2012.
>
> [3] P. K. Agarwal, A. Efrat, and M. Sharir. "Vertical decomposition of shallow levels in 3-dimensional arrangements and its applications." In Proceedings of the eleventh annual symposium on Computational geometry, 1995.
>
> [4] R. Sharathkumar. "A sub-quadratic algorithm for bipartite matching of planar points with bounded integer coordinates." In Proceedings of the twenty-ninth annual symposium on Computational geometry, 2013.
>
> [5] Gabow, Harold N., and Robert E. Tarjan. "Faster scaling algorithms for network problems." SIAM Journal on Computing 18, no. 5 (1989): 1013-1036.
>
> [6] Gabow, Harold N., and Robert E. Tarjan. "Faster scaling algorithms for general graph matching problems." Journal of the ACM (JACM) 38, no. 4 (1991): 815-853.
>
> [7] Lipton, Richard J., and Robert Endre Tarjan. "Applications of a planar separator theorem." In 18th Annual Symposium on Foundations of Computer Science (sfcs 1977), pp. 162-170. IEEE, 1977.

---

> > ### Comment · Reviewer_SwWP · 2023-08-15
> >
> > Thanks authors for your response! I believe in my initial reading of the comparison to the work of Sharathkumar, I didn't realize there were two different sets of papers (Sharathkumar; Sharathkumar and Agarwal) which confused me in understanding how your work built on these past works. I see now this was a misreading on my part.
> >
> > I don't believe I will be updating my score (I still believe this paper should be accepted, and that seems in line with basically all of the other reviewers' assessments), but I will continue to monitor the discussion and update my score if need be!

---

### Official Review · Reviewer_FnYg · 2023-07-02

**Soundness:** 4 excellent
**Presentation:** 4 excellent
**Contribution:** 3 good
**Rating:** 7
**Confidence:** 3

**Summary:**

This paper proposes a new, exact algorithm for solving the Euclidean weighted bipartite matching problem. Here, we have data sets $A,B \subset \mathbb{R}^d,$ each of cardinality $n,$ and the weight of an edge $ab$ is defined to be $\lVert a - b \rVert^p$ for any integer $p\ge 1.$ This formulation is motivated by its direct application to computing empirical $p$-Wasserstein distances.

For data drawn i.i.d. from the same unknown distribution on the unit hypercube, the expected runtime of the algorithm is shown to be weakly-polynomial time, asymptotically in the number of points $n$ and with respect to an additional \emph{spread} parameter $\Delta $, defined to be the ratio of largest and smallest distances between any two points. As $d \to \infty $, this expected runtime approaches that of the classical Hungarian algorithm, up to polylog factors and assuming an efficient data structure for weighted nearest neighbors.

Most of the analysis and all of the experiments focus on the special case where $d=2$, where the improvements over the Hungarian algorithm are more pronounced. The runtime analysis rests on the key observation of Lemma 2.1: if we cut a randomly-shifted square containing the data into four equal pieces, an asymptotically small number of data points will lie near the boundary where these pieces meet. A maximum-weight matching may be constructed by repeated subdivision: optimal matchings in each of the four pieces can be combined to get a feasible solution, and the algorithm searches for "admissible" augmenting paths until the feasible solution can no longer be improved. For a bipartite graph on $2n$ vertices drawn iid unformly from the unit square, the expected runtime of this algorithm is $\tilde O (n^{7/4} \log \Delta )$. For comparison, the Hungarian algorithm in this setting is $\tilde O (n^2)$. Experiments on real and synthetic data show that the new algorithm can substantially outperform the Hungarian algorithm.

Post-rebuttal edit: The authors have responded in a satisfactory manner to my queries. Their algorithm is an interesting addition to the arsenal of matching algorithms, but nevertheless has certain limitations such as dependence on the parameter $\Delta $. For this reason I maintain my high rating of accept.

**Strengths:**

Matching problems are foundational to computer science and optimization. This paper proposes a solution that demonstrates both significant theoretical and empirical advantages in the particular domain of empirical $p$-Wasserstein distance where these problems have been successfully applied. In the class of work proposing improvements to the Hungarian algorithm, the authors have overcome limitations in this domain have included focus on the unweighted case, distributional assumptions, approximate vs exact algorithms, and special cases of the geometric setting studied here.

The runtime analysis is presented very clearly, and theorems are stated carefully. Although the algorithm is ultimately only weakly polynomial, the authors theoretical assertions that the algorithm outperforms the Hungarian method for a range of parameters are backed up convincingly by empirical results.

Another noteworthy feature of the algorithm is that it does not rely on sophisticated data structures like prior work. Indeed, hardly any background at all is needed to understand the proofs and implement the algorithm. This is also reflected in the simplicity of the source code which the authors have shared to aid in reproducibility.

**Weaknesses:**

My main criticism, with a view towards the proposed application of $p$-Wasserstein distance computation, is that the case of _unequal_ distributions is treated mostly as an afterthought. Arguably, this is a more important case in practice since it involves less restrictive assumptions on the data. One finds a few comments addressing this scattered throughout the paper, eg. in the abstract and Remark 3.2. However, to justify the claim that the algorithm performs similarly to the Hungarian algorithm in such cases, it would be beneficial to give experimental results in addition to the asymptotic (as the authors have already done in the equal-distributions case.)

I suggest generally that the authors be somewhat more explicit about the limitations of their algorithm. As already noted in the introduction, computing the empirical Wasserstein distance is mainly tractable in low dimensions. Nevertheless, other readers might be interested in the high-dimensional Euclidean matching problem. I find it difficult to believe that the performance of this method would be better than the purely-combinatorial Hungarian for $d$ large, particularly because of the high cost of dividing each sub-hypercube into $2^d$ pieces.

**Questions:**

line 46: " The execution time of our algorithm is similar to that..." I assume Remark 3.2 is what justifies this remark? It would be helpful to include a forward reference here.

line 59: "and their fast implementations" Don't you mean "its fast implementations"?

line 322: "The dataset consists of the locations" This means latitude and longitude, correct?

line 323: "We filtered the datasets by considering trips" Can you comment more on the choices behind your data filtering and other aspects of experimental design? Specifically, could you report some summary statistics (min, max, median, mean) for the spread of these data sets, both with and without the filtering? It is understandable that some filtering would be needed to see a performance increase over the Hungarian algorithm, and I think the thresholds are reasonable. Still, it would be helpful to understand the effect of filtering on both the runtime and the parameters used to analyze runtime. Moreover, given the filtering step, I think that the suggested improvement $\tilde O( n^{.55})$ is very speculative.

**Limitations:**

For scientific limitations, refer to my comments under "Weaknesses". I do not see any potential for these results to have direct adverse effects on society.

---

> ### Author Rebuttal · Authors · 2023-08-09
>
> We appreciate your thoughtful review and constructive feedback. We address your concerns below.
>
> > My main criticism, with a view towards the proposed application of $p$-Wasserstein distance computation, is that the case of unequal distributions is treated mostly as an afterthought. Arguably, this is a more important case in practice since it involves less restrictive assumptions on the data. To justify the claim that the algorithm performs similarly to the Hungarian algorithm in such cases, it would be beneficial to give experimental results in addition to the asymptotic (as the authors have already done in the equal-distributions case.)
>
> **Response.** Thank you for this question. The case of *equal*  distribution also has significant applications in ML. We have highlighted some of these applications in our common response.
>
> We have also conducted experiments for the unequal case.
> In our experiment on the NY-Taxi data set (already included in our submission), we sampled $n$ drop-off locations and $n$ pick-up locations and match them. Based on our observation, the pick-up and drop-off locations tend to follow different distributions, where pick-ups seem to have a higher density around Manhattan than the drop-offs.
> Furthermore, we conducted additional experiments, where one set is drawn from a Gaussian distribution and the other set is chosen uniformly at random from a unit square. We also included the results of a similar experiment, where one set of points are drawn from the uniform distribution over the unit square and the other set are samples from a Guassian mixture model consisting of $5$ clusters in the $2$-dimensional space. See Figure 1 in the pdf document attached to our general response. We notice an improvement in the efficiency even in these cases.
>
> > I suggest generally that the authors be somewhat more explicit about the limitations of their algorithm. As already noted in the introduction, computing the empirical Wasserstein distance is mainly tractable in low dimensions. Nevertheless, other readers might be interested in the high-dimensional Euclidean matching problem. I find it difficult to believe that the performance of this method would be better than the purely-combinatorial Hungarian for large $d$, particularly because of the high cost of dividing each sub-hypercube into $2^d$ pieces.
>
> **Response.** The divide step of our algorithm takes only $O(dn)$ time.  Indeed, a cell has $2^d$ children, which may be much higher than $n$. However, we note that the only sub-problems that we care about are the ones created by the non-empty children, which are at most $n$, i.e., $O(n)$. One can create these sub-problems by simply scanning the points and placing them in the appropriate sub-problem in $O(dn)$ time.
>
>
> > "We filtered the datasets by considering trips" Can you comment more on the choices behind your data filtering and other aspects of experimental design? Specifically, could you report some summary statistics (min, max, median, mean) for the spread of these data sets, both with and without the filtering? It is understandable that some filtering would be needed to see a performance increase over the Hungarian algorithm, and I think the thresholds are reasonable. Still, it would be helpful to understand the effect of filtering on both the runtime and the parameters used to analyze runtime. Moreover, given the filtering step, I think that the suggested improvement $\tilde{O}(n^{0.55})$ is very speculative.
>
> **Response.** The objective of applying filters is to eliminate erroneous entries in the data, such as entries of trips with negative duration or implausible velocity. To show that the effect of data cleaning step on the result of the experiment was insignificant, we re-executed our algorithm on the NY Taxi dataset, this time applying only two basic filters: (1) the trip duration had to be at least $3$ minutes, and (2) the trip velocity could not exceed $112 mph$. You can see the results in Figure 2 of the PDF file uploaded as part of our general response.

---

> > ### Comment · Reviewer_FnYg · 2023-08-12
> >
> > Thank you for the comments, which have clarified my concerns. I will monitor the discussions before reaching a final decision about the paper rating.

---

### Official Review · Reviewer_NeKH · 2023-07-06

**Soundness:** 3 good
**Presentation:** 3 good
**Contribution:** 3 good
**Rating:** 6
**Confidence:** 3

**Summary:**

The paper considers matching two sets $A$ and $B$ of $n$ points in the Euclidean space so as to minimize the sum of distances of matched points, when both pointsets are drawn independently and identically from the same (unknown to the algorithm) distribution.  The authors extend the well-known Hungarian method with the shifted quad-tree decomposition technique, and improve over the best known result for the worst-case pointsets. The paper is complemented with experimental results.

**Strengths:**

This is an interesting problem to consider and I found the extension of the Hungarian method with the shifted-quad tree technique quite interesting and -- at least to me -- novel. The paper is well written.

**Weaknesses:**

The result of the paper is novel but I am not sure if I would call a better performance for only a special class of inputs an improvement over (slightly worse) results but that hold for general inputs.

I would have preferred if the paper explicitly stated it every time a technique from the literature is used/adapted.

**Questions:**

- I was convinced by your motivation for studying the problem, but I was wondering if you know of any real-world matching problem where it is natural to have the points drawn by the same distribution?

**Limitations:**

No limitations applicable and no potential negative societal impact.

---

> ### Author Rebuttal · Authors · 2023-08-09
>
> We appreciate your thoughtful review and feedback.
>
> >  The result of the paper is novel but I am not sure if I would call a better performance for only a special class of inputs an improvement over (slightly worse) results but that hold for general inputs.
>
> **Response.** We do not claim a better performance but a more *robust* performance in comparison to the Hungarian algorithm. Our algorithm has a similar worst-case performance to the Hungarian method but a faster performance for stochastic point sets (see our general response for a discussion on this).
>
> > I was convinced by your motivation for studying the problem, but I was wondering if you know of any real-world matching problem where it is natural to have the points drawn by the same distribution?
>
> **Response.** There are many ML problems that require testing if two sample sets are from the same distribution.
>
> * Distributional shifts: Does the real data set represent the same distribution as the training data from which ML models are built? [1]
> * Benchmarks: Do models built using different ML techniques on the same training data represent the same distribution? [2, 3]
> * Mutual-independence: Given a multi-variate distribution, are the two marginals of the distribution independent? [4]
>
> The questions above reduce to the *two-sample test problem*: Given two sets of $n$ samples, determine if they are drawn from the same multivariate distribution or different ones. In the *Wasserstein two sample test*, one checks if the optimal matching cost between the samples is below a threshold to determine if they are from the same distribution [5]. The computation of the optimal matching cost can be done using our algorithm.
>
> ---
> ---
>
> **References.**
>
> [1] S. Rabanser, S. Günnemann, and Z. Lipton. "Failing loudly: An empirical study of methods for detecting dataset shift." Advances in Neural Information Processing Systems, 2019.
>
> [2] A. Borji. "Pros and cons of gan evaluation measures." Computer vision and image understanding, 2019.
>
> [3] D. Lopez-Paz, and M. Oquab. "Revisiting classifier two-sample tests." arXiv preprint, 2016.
>
> [4] N. Deb, and B. Sen. "Multivariate rank-based distribution-free nonparametric testing using measure transportation." Journal of the American Statistical Association, 2023.
>
> [5] M. Imaizumi, H. Ota, and T. Hamaguchi. "Hypothesis Test and Confidence Analysis With Wasserstein Distance on General Dimension." Neural Computation, 2022.

---

> > ### Comment · Reviewer_NeKH · 2023-08-11
> >
> > Thanks a lot for the response and the clarifications. My evaluation remains unchanged.

---

### Author Rebuttal · Authors · 2023-08-09

Thank you for the very positive feedback on our work. We want to emphasize a few important points that were also presented in our manuscript and hope that these points may help address some of the reviewers' criticisms.

**Novelty.** The novelty of our algorithm is in placing the classical Hungarian algorithm within a geometric divide-and-conquer framework. Hungarian algorithm conducts a "global Hungarian search" to match each point. In contrast, our algorithm obtains speed-up by trapping shorter edges of the optimal matching in smaller sub-problems (squares) of the quadtree and matching these points using a Hungarian search that is local to the sub-problem. While our asymptotic improvements are shown for stochastic points that are drawn from the same distribution, we do expect our algorithm to perform better when the optimal matching has many edges with small cost, for instance, when points are drawn from two similar but unequal distributions. This is best exemplified by our experiment on NY-Taxi data set (already included in our submission), where we sample $n$ drop-off locations and $n$ pick-up locations and match them. Based on our observation, the pick-up and drop-off locations tend to follow different distributions, where pick-ups seem to have a higher density around Manhattan than the drop-offs. We also conducted two additional experiments on samples drawn from two different distributions (the first using Gaussian and uniform distributions, and the second using a Gaussian mixture model with $5$ clusters and a uniform distribution) and include the results in the one-page pdf submitted as part of our response.

In terms of techniques, our algorithm does not rely on any prior work except for a weighted nearest-neighbor based implementation of the Hungarian search step (which was proposed in [1, 2] and used in [3, 4]). Our algorithm also uses a randomly shifted quadtree, which is a popular data structure for the design of geometric approximation algorithms. To the best of our knowledge, our algorithm is the first to use them in the design of an exact Euclidean bipartite matching algorithm.

**Applications.** There are many ML problems that require testing if two sample sets are from the same distribution.

* Distributional shifts: Does the real data set represent the same distribution as the training data from which ML models are built? [5]
* Benchmarks: Do models built using different ML techniques on the same training data represent the same distribution? [6, 7]
* Mutual-independence: Given a multi-variate distribution, are the two marginals of the distribution independent? [8]

The questions above reduce to the *two-sample test problem*: Given two sets of $n$ samples, determine if they are drawn from the same multivariate distribution or different ones. In the *Wasserstein two sample test*, one checks if the optimal matching cost between the samples is below a threshold to determine if they are from the same distribution [9]. The computation of the optimal matching cost can be done using our algorithm.
Two-sample testing dates back to Leighton and Shor [10], who used it to evaluate the quality of pseudo-random number generators.

Several ML problems generate samples from similar (but not identical) distributions, such as the domain adaptation problem [11] and the training of a Wasserstein GAN [12]. In both instances, our algorithm can be faster than Hungarian when the generated samples are low-dimensional.

---
---
**References.**

[1] P. Vaidya. "Geometry helps in matching." In Proceedings of the twentieth annual ACM symposium on theory of computing, 1988.

[2] R. Sharathkumar, and P. K. Agarwal. "Algorithms for the transportation problem in geometric settings." In Proceedings of the twenty-third annual ACM-SIAM symposium on Discrete Algorithms, 2012.

[3] P. K. Agarwal, A. Efrat, and M. Sharir. "Vertical decomposition of shallow levels in 3-dimensional arrangements and its applications." In Proceedings of the eleventh annual symposium on Computational geometry, 1995.

[4] R. Sharathkumar. "A sub-quadratic algorithm for bipartite matching of planar points with bounded integer coordinates." In Proceedings of the twenty-ninth annual symposium on Computational geometry, 2013.

[5] S. Rabanser, S. Günnemann, and Z. Lipton. "Failing loudly: An empirical study of methods for detecting dataset shift." Advances in Neural Information Processing Systems, 2019.

[6] A. Borji. "Pros and cons of gan evaluation measures." Computer vision and image understanding, 2019.

[7] D. Lopez-Paz, and M. Oquab. "Revisiting classifier two-sample tests." arXiv preprint, 2016.

[8] N. Deb, and B. Sen. "Multivariate rank-based distribution-free nonparametric testing using measure transportation." Journal of the American Statistical Association, 2023.

[9] M. Imaizumi, H. Ota, and T. Hamaguchi. "Hypothesis Test and Confidence Analysis With Wasserstein Distance on General Dimension." Neural Computation, 2022.

[10] F. T. Leighton, and P. Shor. "Tight bounds for minimax grid matching, with applications to the average case analysis of algorithms." In Proceedings of the eighteenth Annual ACM symposium on theory of computing, 1986.

[11] Y. Balaji, R. Chellappa, and S. Feizi. "Robust optimal transport with applications in generative modeling and domain adaptation." NeurIPS, 2020.

[12] H. Liu, G. U. Xianfeng, and D. Samaras. "A two-step computation of the exact gan wasserstein distance." In ICML, 2018.

---

### Decision · Program_Chairs · 2023-09-21

**Decision:**

Accept (poster)

**Comment:**

This paper reduces the runtime complexity of bipartite matching, when the points in the two partitions are drawn independently and random from unknown distribution. The authors discuss some applications like NY-Taxi data set (already included in our submission), where they sample drop-off locations and pick-up locations and match them.